# Multiple golgins are required to support extracellular matrix secretion, modification, and assembly

George Thompson[1], Anna Hoyle[2,3], Philip A. Lewis[4], M. Esther Prada-Sanchez[1], Joe Swift[2,3], Kate Heesom[4], Martin Lowe[5], David Stephens[1], and Nicola L. Stevenson[1]

The secretion of extracellular matrix (ECM) proteins is vital to the maintenance of tissue health. One major control point of this process is the Golgi apparatus, whose dysfunction causes numerous connective tissue disorders. We therefore sought to investigate the role of two Golgi organizing proteins, GMAP210 and Golgin-160, in ECM secretion. CRISPR knockout of either protein had distinct impacts on Golgi organization, with Golgin-160 knockout causing Golgi fragmentation and vesicle accumulation, and GMAP210 loss leading to cisternal fragmentation, dilation, and the accumulation of tubulovesicular structures. Both golgins were required for fibrillar collagen organization and glycosaminoglycan synthesis suggesting nonredundant functions in these processes. Furthermore, proteomics analysis revealed both shared and golgin-specific changes in the secretion of ECM proteins. We therefore propose that golgins are collectively required to create the correct physical–chemical space to support efficient ECM protein secretion, modification, and assembly. This is the first time that Golgin-160 has been shown to be required for ECM secretion.

## Introduction

The extracellular matrix (ECM) is a complex network of proteins, carbohydrates, and minerals secreted by cells to build a scaffold that forms the structural basis of most tissues. To date, current resources describing the "matrisome" have identified >1,000 proteins associated with this compartment in humans (Shao et al., 2023), explaining the capacity for extensive diversification of biochemical and mechanical properties between tissue types. The most abundant family of ECM proteins are the collagens. These proteins assemble to form the main structural components of connective tissues alongside other fiber- and network-forming proteins such as fibronectin and elastin (Karamanos et al., 2021). This assembly is supported by accessory factors like the proteoglycans (Couchman and Pataki, 2012), a diverse set of proteins characterized by the presence of at least one glycosaminoglycan (GAG) chain that helps to promote tissue hydration and sequester signaling molecules (Gandhi and Mancera, 2008). Meanwhile, ECM turnover is regulated by secreted proteases (Arpino et al., 2015; Lee and Murphy, 2004).

All newly synthesized ECM proteins must transit the secretory pathway to reach the extracellular environment for assembly. As they do so, many are heavily modified, which alters their chemistry and consequently the way in which they interact and assemble (Adams, 2023). Regulation of ECM protein modification is therefore crucial to the assembly of a functional matrix. One of the main compartments responsible for facilitating this process is the Golgi apparatus, which contains enzymes catalyzing modifications such as O-glycosylation, N- and O-linked glycan maturation, proteolytic processing, phosphorylation, lipidation, and sulfation (Potelle et al., 2015). Given this central role in protein processing, it is unsurprising that a large number of genes affecting Golgi function have been linked to various connective tissue disorders (Hellicar et al., 2022).

Golgi structure is highly complex and tightly linked to its function. In mammalian systems, it is composed of a stack of flattened, fenestrated membranous compartments called cisternae, which are laterally connected to form an interconnected

[1]School of Biochemistry, University of Bristol, Bristol, England; [2]Wellcome Centre for Cell-Matrix Research, Manchester, UK; [3]Division of Cell Matrix Biology and Regenerative Medicine, School of Biological Sciences, Faculty of Biology, Medicine and Health, Manchester Academic Health Science Centre, University of Manchester, Manchester, UK; [4]Proteomics Facility, University of Bristol, Bristol, England; [5]School of Biological Sciences, Faculty of Biology, Medicine and Health, University of Manchester, Manchester, UK.

Professor Stephens died on October 15, 2024.   Correspondence to Nicola L. Stevenson: nicola.stevenson@bristol.ac.uk

A. Hoyle's current affiliation is Kennedy Institute of Rheumatology, University of Oxford, Oxford, UK.

Golgi ribbon (Lowe, 2011). These stacks are polarized such that cargoes arriving from the ER enter the Golgi stack at the *cis*-face, transit through the *medial*- and *trans*-compartments, and then get sorted before exiting the Golgi at the *trans*-Golgi network (TGN). Golgi-resident enzymes are distributed across the cisternae with a *cis-trans* polarity to support the successive addition of post-translational modifications to cargoes as they transit the compartments (Rabouille et al., 1995; Tie et al., 2022). This distribution is maintained by intra-Golgi vesicular transport, which recycles enzymes between Golgi cisternae (Arab et al., 2024).

One of the major protein families responsible for maintaining Golgi organization is the golgin family (Barr and Short, 2003; Gillingham and Munro, 2016; Witkos and Lowe, 2015). These proteins project coiled-coil domains into the cytosol from the Golgi surface to tether other Golgi membranes and transport vesicles (Lowe, 2019; Wong and Munro, 2014). Specific golgins localize to distinct regions of the Golgi organelle, and this, combined with their affinity for different vesicles, confers directionality and specificity to intra-Golgi traffic (Wong et al., 2017). We and others have shown that one of these golgins, giantin, is required for the secretion of a healthy ECM. Loss of giantin prevents proper processing of the procollagen type I N-propeptide (Stevenson et al., 2021), impacts the abundance of ECM proteins (Katayama et al., 2018; Kikukawa and Suzuki, 1992), and impairs GAG metabolism (Kikukawa et al., 1991a; Kikukawa et al., 1991b; Kikukawa et al., 1990). These phenotypes are variable between animal models, but, interestingly, always impact skeletal development (Katayama et al., 2011; Lan et al., 2016; Stevenson et al., 2017; Suzuki et al., 1988).

Another golgin that has been linked to ECM deposition and skeletal development is GMAP210. Mutations in the *TRIP11* gene encoding GMAP210 are causative of the human skeletal diseases achondrogenesis type 1A and odontochondrodysplasia (Costantini et al., 2021; Del Pino et al., 2021; Medina et al., 2020; Qian et al., 2021; Smits et al., 2010; Upadhyai et al., 2021; Wehrle et al., 2019; Yeter et al., 2022). Interestingly, *Trip11* mutant mice have a similar phenotype to giantin mutant rats, with both models showing neonatal lethal chondrodysplasia characterized by craniofacial defects, shortened limbs and ribs, and delayed mineralization of bone (Bird et al., 2018; Follit et al., 2008; Smits et al., 2010; Yamaguchi et al., 2021). Loss of GMAP210 also impacts the secretion of specific ECM proteins in a selective manner (Bird et al., 2018; Smits et al., 2010; Yamaguchi et al., 2021).

Altogether, these studies suggest that ECM secretion is particularly susceptible to the loss of at least two golgins, which appear to act nonredundantly in this context. This led us to question the extent to which ECM deposition is sensitive to the loss of golgins more widely, and how this relates to changes in Golgi organization. To begin addressing this, we performed a comparative study between GMAP210 and a third golgin, Golgin-160. Golgin-160 was chosen because it resides on *cis*-Golgi membranes like GMAP210 (Hicks et al., 2006) but remains poorly characterized at a functional level. Here, we report that knockout (KO) of Golgin-160 has a similar impact on ECM organization as GMAP210 loss. Golgi morphology, on the other hand, is affected differently, consistent with the two golgins acting in different ways to maintain Golgi homeostasis. We therefore conclude that GMAP210 and Golgin-160 function nonredundantly to create the correct physical and biochemical space to support the modification and secretion of complex ECM molecules.

## Results

### Generation of golgin KO cell lines

To test the role of different golgins on matrix secretion, we generated two GMAP210 and Golgin-160 KO clones using CRISPR/Cas9 genome engineering in human retinal pigment epithelial cells (RPE1). GMAP210 KO clone 1 was generated using a gRNA targeting exon 4 of the *TRIP11* gene (Fig. S1 A). This exon is common to both predicted protein-coding transcripts TRIP11-201 (Ensembl transcript ID ENST00000267622.8, UniProt ID Q15643-1) and TRIP11-202 (ENST00000554357.5, H0YJ97), and so mutation should impact all protein expression. The generated mutation was a single base pair insertion of alanine resulting in a frameshift from position Glu[114] and truncation at amino acid 155 (Fig. S1 A). GMAP210 KO clone 2 was generated using a gRNA targeting exon 1 of transcript TRIP11-202 and upstream of the predicted start site of the longer transcript TRIP11-201. This resulted in a 10-bp deletion in the promoter region (Fig. S1 A). Loss of protein expression was confirmed in both clones by immunofluorescence using three different antibodies targeting the N and C terminus of the GMAP210 protein (Fig. S1 B) and western blotting (Fig. S1 C). Note trace amounts of protein could be detected with the C-terminal antibody, suggesting a truncation product may persist. Nonetheless, we considered this sufficient depletion for further study.

Golgin-160 KO clones were generated using a gRNA targeting exon 20 of transcript GOLGA3-201 (ensemble ID ENST00000204726.9, UniProt ID: Q08378-1). This is expected to mutate all predicted transcripts encoding fragments larger than 44 amino acids. Mutation with this gRNA consistently produced a single alanine insertion, resulting in frameshift at amino acid Glu[1312] and truncation at amino acid 1331 (Fig. S1 A). Consequently, two clones with the same mutation were carried forward for experiments to rule out clonal effects. Protein loss was confirmed by immunofluorescence and western blotting, again with antibodies targeting both the N and C terminus (Fig. S1, D and E).

### Loss of either GMAP210 or Golgin-160 causes changes in ECM assembly and organization

To investigate the requirement for GMAP210 and Golgin-160 in ECM secretion and assembly, we grew our new KO lines for a week in the presence of ascorbic acid to promote collagen secretion and visualized the cell-derived ECM by immunofluorescence labeling. Co-labeling of two major fibrillar ECM components, collagen type 1 and fibronectin, revealed an extensive network of highly aligned and organized extracellular fibrils in wild-type (WT) cell cultures, with cell nuclei stretching along the fibril axis (Fig. 1, A–D). In GMAP210 (Fig. 1, A and B) and Golgin-160 KO (Fig. 1, C and D) cell cultures on the other hand, fibrils were identifiable, but they were less well ordered, showing poor alignment and an increase in branching. To

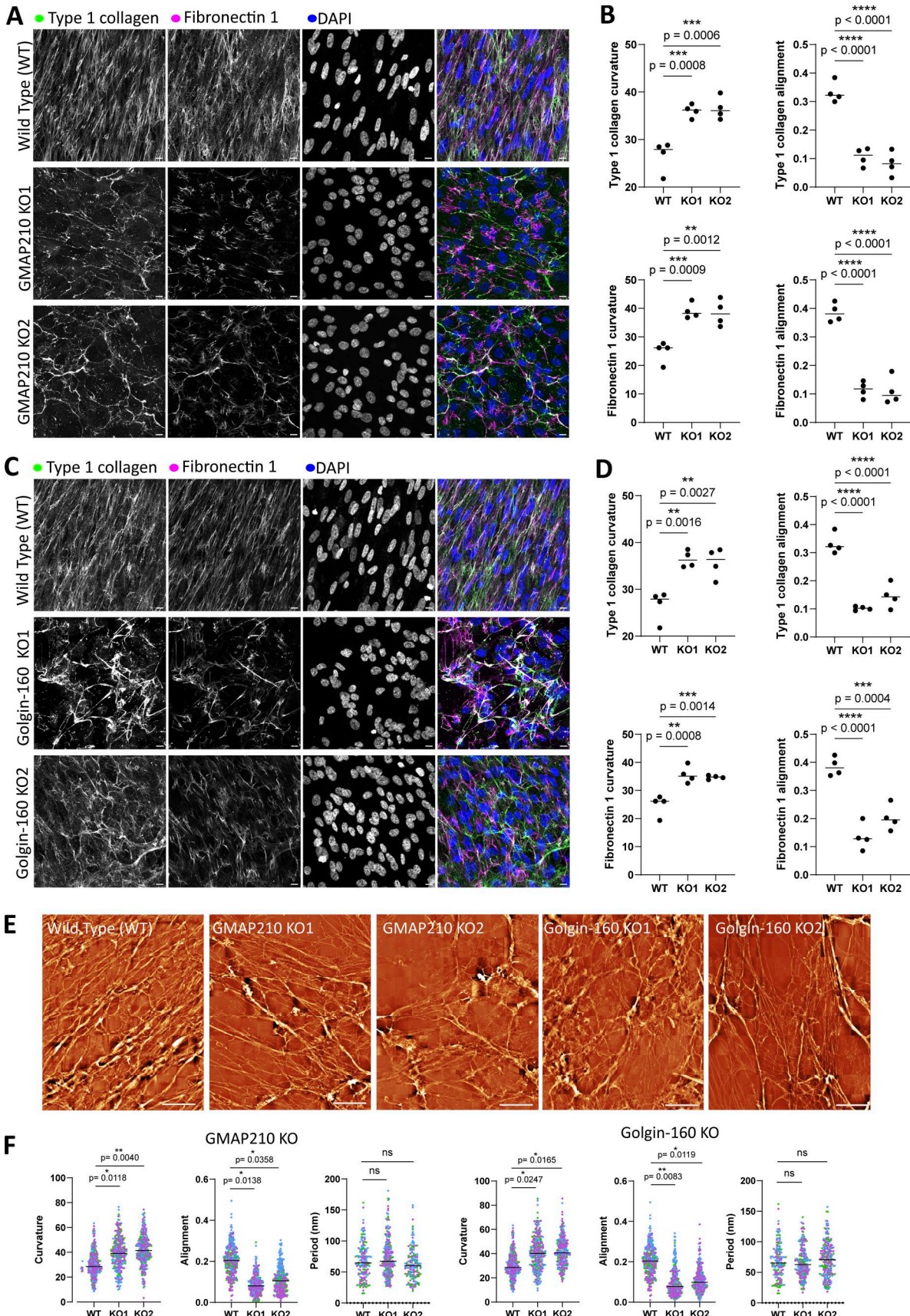

Figure 1. **ECM organization is altered in golgin mutant cultures. (A and C)** Confocal maximum projection images of non-permeabilized WT, GMAP210 KO (A), and Golgin-160 KO (C) RPE1 cell cultures immunolabeled for extracellular collagen type I (green), fibronectin-1 (magenta), and nuclei (DAPI, blue). Scale

bars, 10 µm. **(B and D)** Quantification of fibril characteristics measured from images represented in A and C using the TWOMBLI ImageJ plug-in (see Materials and methods). Individual dots represent the mean of each biological replicate (*n* = 4), and bars represent the median of all experiments. **(E)** Decellularized ECM from WT, GMAP210 KO, and Golgin-160 KO cell cultures imaged by HS-AFM. Images are representative of 10 × 10 raster scans from biological replicates (*n* = 3). Scale bars, 10 µm. **(F)** TWOMBLI quantification of fibril characteristics measured from images represented in E. Individual dots are measurements from each tile in the raster scan, with each biological replicate color-coded (*n* = 4). Bars represent the median of all experiments. **(B, D, and F)** Data were subjected to a Shapiro–Wilk test for normality and then a nested one-way ANOVA with Dunnett's test for multiple comparisons to generate P values. ANOVA, analysis of variance.

---

examine the structure of these fibrils more closely, the cell layer was extracted, and the remaining cell-derived matrix was imaged by high-speed atomic force microscopy (HS-AFM). Again, fibrils were less well organized in all golgin KO cultures (Fig. 1, E and F). They were also thinner; however, the periodicity of the collagen fibrils was largely unchanged, suggesting assembly of collagen trimers into fibrils is normal, but lateral interconnections between fibrils may be reduced.

The overall fluorescence intensity of collagen staining in the matrix appeared reduced in the KO cultures. We therefore determined the abundance of collagen type I in lysate and matrix fractions of cell cultures by immunoblotting against the Col1a1 chain. Col1a1 abundance in the lysate fractions of mutant cultures was comparable to that of WT lysates when normalized to total protein, suggesting expression is normal and collagen is not retained inside the cell (Fig. 2, A and C i). Col1a1 abundance was, however, reduced in the ECM fraction of all mutant lines when normalized against total cellular protein (Fig. 2, B and C ii), suggesting it is not being efficiently incorporated into the insoluble matrix. Collagen in the media was below the level of detection by this method.

## ECM composition is altered following loss of GMAP210 and Golgin-160

To determine whether the observed changes in ECM organization in the golgin KO cultures reflect changes in ECM composition, the cell-derived matrix from WT and KO cultures was collected and analyzed by mass spectrometry. Principal component analysis showed distinction between the WT and Golgin-160 KO and between the KO matrix samples, but variation between repeats was high (Fig. S2 A). Results were filtered against the matrisome database (Shao et al., 2023) to indicate proteins specifically related to ECM structure and function. Overall, fewer proteins were significantly impacted by loss of Golgin-160 than of GMAP210; however, there was a great deal of overlap between the KO lines (Fig. 3). For example, laminin subunit γ 1 (LAMC1), laminin subunit α 5 (LAMA5), hemicentin-1 (HMCN1), latent transforming growth factor β binding protein 1 (LTBP1), and transforming growth factor β 2 (TGFB2) were significantly reduced in abundance in at least one Golgin-160 KO line, as well as the GMAP210 KOs (Fig. 3, A–C). Conversely, the abundance of fibrillins was increased in all KO lines (Fig. 3, A–C). Interestingly, lysyl oxidase like 1 (LOXL1), which was upregulated in the GMAP210 KO ECM,

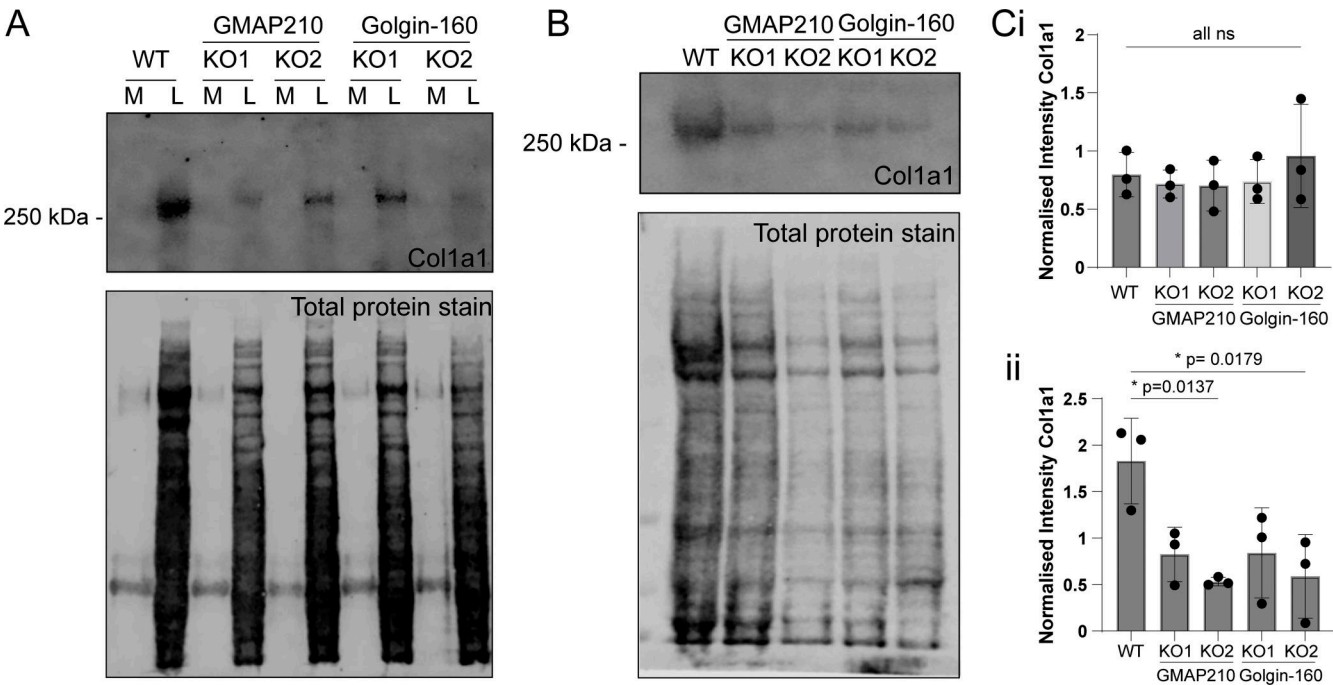

**Figure 2.** **Collagen deposition in the matrix is impaired in golgin mutant cells. (A)** Immunoblot for Col1a1 and total protein stain after SDS-PAGE of medium (M) and lysate (L) samples taken from WT and golgin KO cultures. **(B)** Immunoblot for Col1a1 and total protein stain after SDS-PAGE of the cell-derived matrix extracted from WT and golgin mutant cultures. **(C i and ii)** Quantification of Col1a1 intensity normalized against total cellular protein for i lysate samples as represented in A and ii matrix samples as represented in B. Dots show individual experiment result (*n* = 3), and bars show the mean and standard deviation. Data were subjected to a Shapiro–Wilk test for normality and then a nested one-way ANOVA with Dunnett's test for multiple comparisons to generate P values. Source data are available for this figure: SourceData F2.

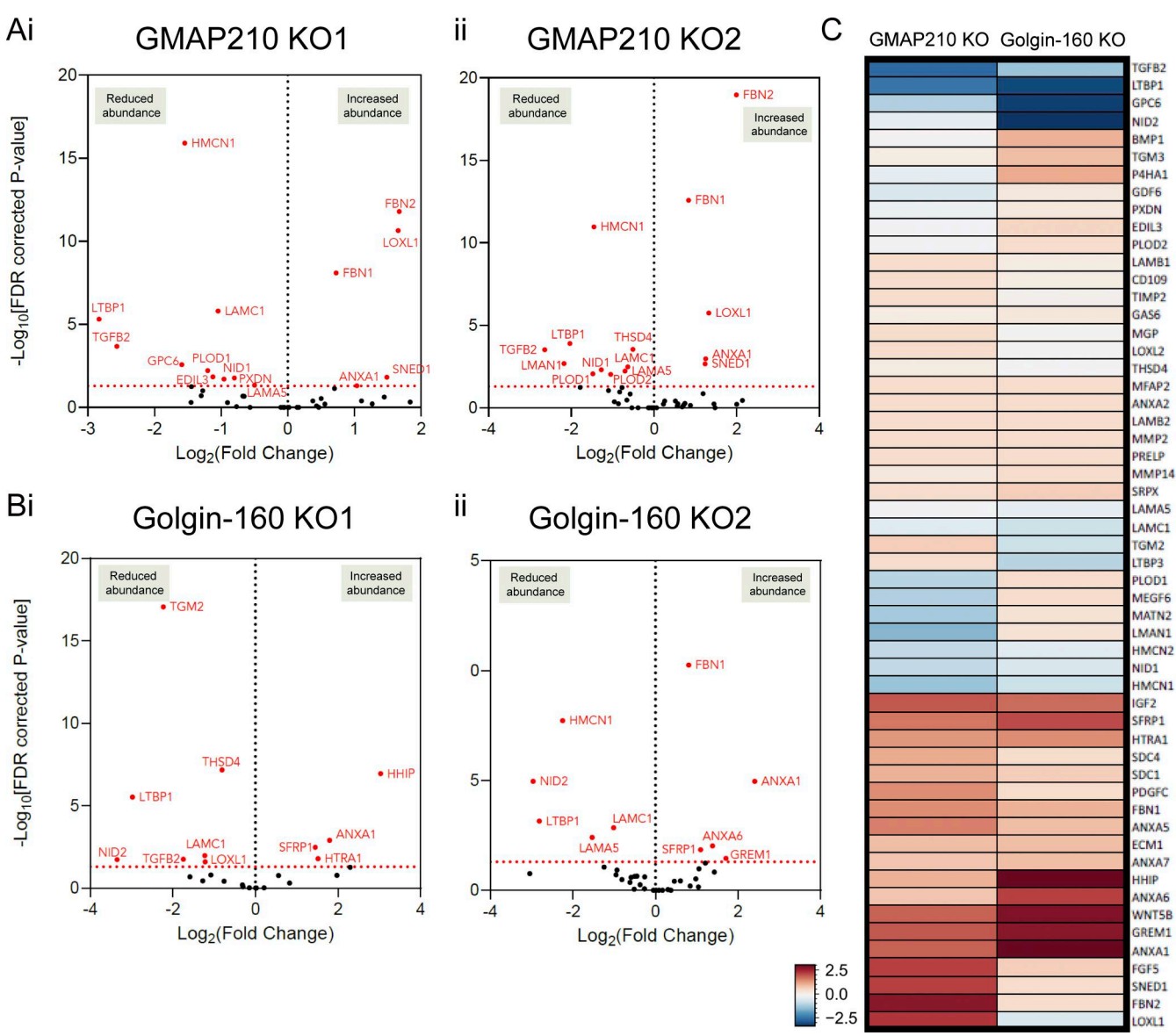

Figure 3. **ECM composition is altered in golgin KO cells. (A and B)** Volcano plots showing log-fold change in cell-derived matrix protein abundance from GMAP210 KO (A) or Golgin-160 KO (B) cells compared with WT. Red-labeled points represent proteins with significantly changing abundance, with the Benjamini–Hochberg FDR-corrected P values <0.05, *n* = 3. Only proteins categorized as matrisome are represented on these plots (Shao et al., 2023). **(i and ii)** Results for two different clones per gene are shown. **(C)** Heat map comparing protein abundance changes between the two different golgin KOs after pooling both clones. Red and blue indicate increased and decreased abundance respectively. Color intensity is determined by the average log-fold change across mutants compared with WT.

was downregulated in Golgin-160 KOs, and vice versa for procollagen-lysine,2-oxoglutarate 5-dioxygenase 1 (PLOD1, Fig. 3, A–C). This hints at golgin-specific changes in the post-translational modification of collagen. Altogether, these data show that both GMAP210 and Golgin-160 are required for the deposition of a well-organized ECM. The similarity seen between ECM phenotypes in GMAP210 and Golgin-160 KO cells also demonstrates the non-redundant nature of these golgins in this context and the broad susceptibility of ECM cargoes to golgin loss.

**GMAP210 and Golgin-160 are not required for procollagen processing**
We have shown previously that KO of giantin disrupts procollagen processing (Stevenson et al., 2021). To test whether

impaired collagen processing is contributing to the ECM phenotypes reported here too, we interrogated our ECM proteomics data set for peptides relating to collagen prodomains. Collagen type IV had the highest peptide count and therefore provided the most robust analysis. Comparing the average peptide counts across the Col4a2 sequence, we found no significant increase in the number of peptides containing an intact N-propeptide cleavage site in the KO cultures (N-propeptide overlap sequence, Fig. S2, B and C), indicating that processing of collagen type IV proceeds effectively in the absence of Golgin-160 and GMAP210. The expression of a procollagen type 1 construct in which the N-propeptide is GFP-tagged (McCaughey et al., 2019) also revealed the presence of free collagen type I N-propeptide cleavage products in both WT and Golgin-160 KO cells indicating

successful cleavage (Fig. S2 D). GMAP210 KO cells did not tolerate procollagen type I overexpression and could not be tested. Altogether, these data indicate that loss of GMAP210 or Golgin-160 does not phenocopy loss of giantin with respect to collagen processing and that this is not a contributing factor to the observed ECM phenotypes here.

## Loss of GMAP210 or Golgin-160 leads to distinct changes in Golgi morphology

Having established that loss of GMAP210 or Golgin-160 has a similar impact on ECM deposition, we next sought to determine the impact of golgin loss on Golgi organization. Cells were labeled with markers of the *cis*- and *medial*-Golgi and the TGN. Golgi area and extent of fragmentation were then measured (Fig. 4, A–D). Strikingly, each golgin KO had a different impact on Golgi morphology. As previously reported (Bird et al., 2018; Sato et al., 2015; Smits et al., 2010; Wehrle et al., 2019; Yamaguchi et al., 2022), loss of GMAP210 resulted in a clear compaction of Golgi structures compared with WT cells (Fig. 4, A and B). The Golgi also appeared fragmented, although this was difficult to quantify due to the compaction and poor definition of Golgi elements. In Golgin-160 KO cells on the other hand, no consistent change in Golgi size was observed, but there was a clear increase in Golgi fragment number (Fig. 4, C and D). In all KO lines, the *cis-trans* polarity of the Golgi appeared to be maintained, although again this was harder to ascertain in the GMAP210 KO cells (Fig. 4, A and C). These phenotypes were also confirmed in a second cell line, mouse MC3T3-E1, by CRISPR/Cas9 transfection and immunostaining of GM130 to label the Golgi, and Golgin-160 or GMAP210 to identify KO cells (Fig. S3 A).

To better define the impact of KO on Golgi membrane organization, higher resolution imaging was performed using single-tilt electron tomography. For each cell line, 3D tomographic reconstructions were generated (Videos 1, 2, and 3) and a section of the Golgi was segmented to build a 3D model of membrane organization (Videos 1, 2, and 3; and Fig. 4, E–G). In these models, membranes associated with the Golgi apparatus were grouped into four structural categories: cisternae (blue), defined as laterally flattened, stacked membranous compartments; dilated structures (red)—large volume structures lacking flattened regions; tubulovesicular structures (yellow)—convoluted structures with interconnected regions of spherical and tubular shape; and vesicles (green)—small, spherical structures.

In WT cells, as expected, we observed tightly stacked, evenly flattened, fenestrated cisternae that were laterally connected to span lengths of tens of microns (Fig. 4 E). Dilated and tubulovesicular structures were predominantly localized to the *trans*-side of the Golgi in these cells, with small numbers of vesicles concentrated at the cisternal rims and *trans*-Golgi face. In contrast, although the Golgi was identifiable in GMAP210 KO cells, membranes were highly disordered (Fig. 4 F). The cisternae were mostly stacked, but they were short with no lateral connections and few fenestrations, and many were dilated or had bulges along their length, particularly at the *cis*- and *trans*-end of the stack. Some membranous structures resembling these dilated cisternae were also observed near, but not part of, a stack,

but their origin and identity are hard to assign. Finally, in many instances the cisternae were surrounded by extensive tubulovesicular structures interspersed with vesicles that wrapped around the mini-stacks (Fig. 4, F iii and iv). Some of these structures had tunnel-like indentations, which in cross section appeared as internalized vesicles but were continuous with the cytosol (Fig. 4, F ii and iii, open-headed arrows). Note that we did not see any obvious ER swelling as has been reported to occur in a cell type–specific manner in some *in vivo* studies (Bird et al., 2018; Smits et al., 2010; Yamaguchi et al., 2021). Despite the similarity of the tubulovesicular membranes to ERGIC membranes (Appenzeller-Herzog and Hauri, 2006), we did not see any gross changes to ERGIC53 distribution in these cells at the light level (Fig. S3 B). To conclude, the geometry and organization of the Golgi membrane are severely perturbed in the absence of GMAP210 in a way that is suggestive of both structural and trafficking defects.

Examination of Golgin-160 KO cells showed well-stacked, tightly flattened cisternae that remained juxtanuclear as in WT cells (Fig. 4 G). However, consistent with the immunofluorescence data, there were very few lateral connections between cisternae meaning membranes were fragmented into closely apposed mini-stacks rather than forming a classical ribbon architecture (Fig. 4, G i, pink brackets). Cisternae also appeared to have internal vesicles or holes unlike control cells (Fig. 4, G ii, double-headed arrows). Strikingly, there was a clear accumulation of vesicular structures in the area surrounding the cisternal rims, as well as both the *cis*- and *trans*-Golgi face (Fig. 4, G v). This coincided with a high incidence of spherical structures attached to the membranes of cisternal rims that resembled frustrated vesicle budding or fusion profiles (Fig. 4, G ii, pink arrow). Intriguingly, budding vesicles were also frequently observed at the nuclear envelope (Fig. 4, G iii, closed-headed arrows). Although the accumulating vesicles were reminiscent of COPI vesicles with an electron-dense coat, we were unable to observe any gross changes in COPI localization at the light level by immunofluorescence as individual puncta could not be distinguished in the Golgi region (Fig. S3 C). In conclusion, loss of GMAP210 or Golgin-160 has a different impact on Golgi structural organization.

## Loss of either GMAP210 or Golgin-160 leads to reduced secretion of ECM components and regulators

Given the clear disruption to Golgi organization in our KO lines, we decided to look at secretion more generally. Cells were grown in serum-free medium overnight, before collecting the media to analyze the abundance of soluble secreted proteins by mass spectrometry. Three independent replicate samples were collected for all WT and KO lines and analyzed simultaneously by 15-plex tandem mass tagging mass spectrometry. The sum of raw abundance revealed replicate one to have contained more protein, and thus, pairwise comparisons within replicates were chosen over averages (Fig. S4 A). Principal component analysis reiterated the need for paired analysis with PC1 and PC2 separating the samples by replicate; however, good separation was observed at PC3 and PC4 between WT, Golgin-160 KO, and GMAP210 KO lines, with individual clones clustering together

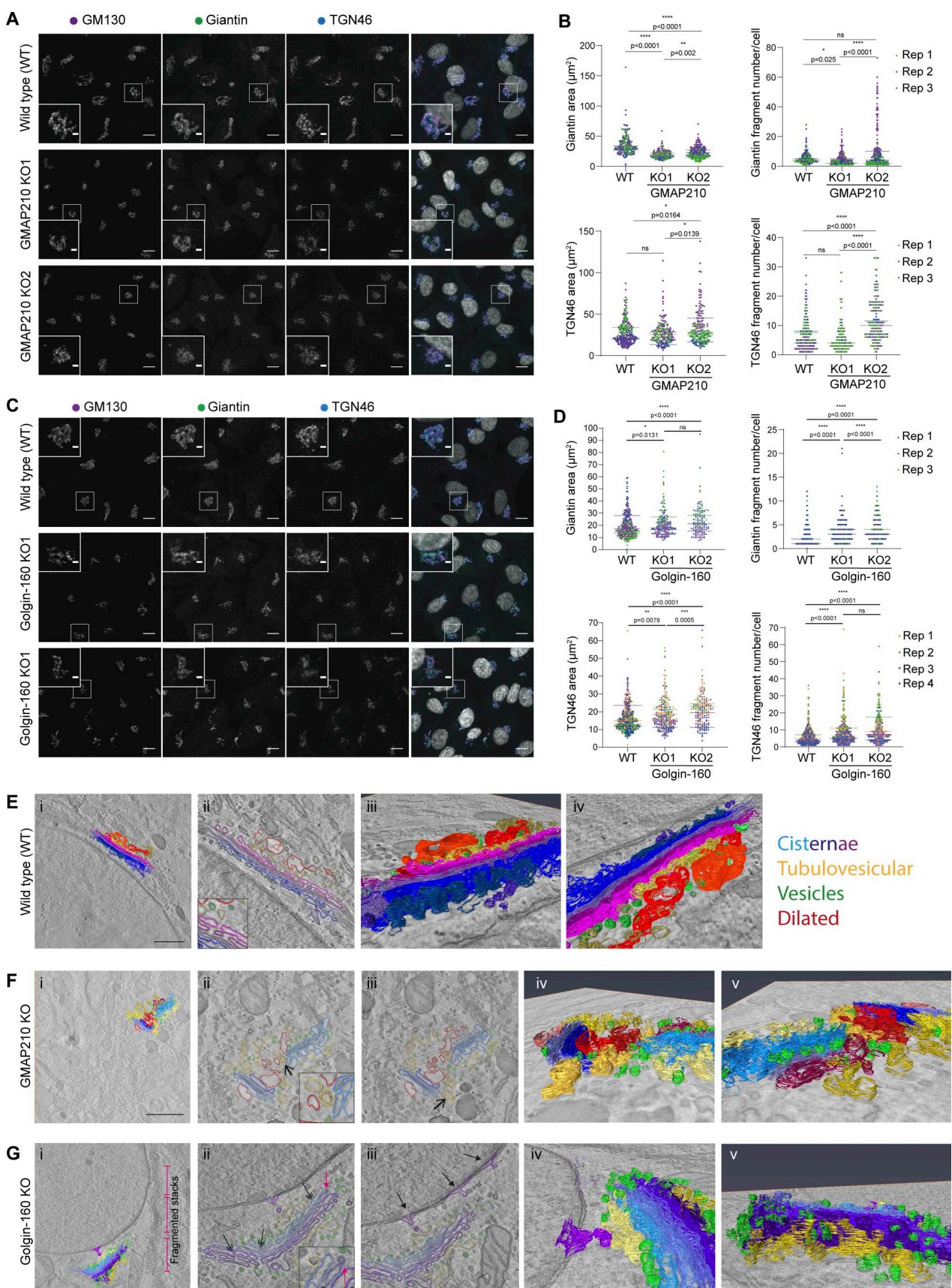

Figure 4. **Golgi organization is altered upon loss of GMAP210 or Golgin-160. (A and C)** Maximum projection confocal images of WT and GMAP210 KO (A) and Golgin-160 KO (C) RPE1 cells immunolabeled for *cis*-Golgi (GM130, magenta), *cis/medial*-Golgi (giantin, green), and TGN (TGN46, blue) markers. Nuclei

labeled with DAPI (grayscale). Scale bar, 10 µm. Inset scale bar, 1 µm. **(B and D)** Quantification of total giantin and TGN46 area and fragment number per cell from images represented in A and C. Individual dots represent one cell and are colored by replicate (n = 3). Bars show the median from each replicate experiment. Statistical analysis was performed using a Shapiro–Wilk normality test and a Kruskal–Wallis significance test. **(E–G)** Tomographic reconstructions of Golgi structures in WT (E), GMAP210 KO (F), and Golgin-160 KO (G) cells. Segmented membranes are labeled as cisternae (blue/purple), dilated structures (red), tubulovesicular structures (yellow), and vesicles (green). **(E ii, F ii, iii, and G ii iii)** Single-slice images with segmentation. **(F ii and iii)** Open arrows point to invaginations within spherical regions of tubulovesicular structures. **(G ii and iii)** Double-headed arrows indicate top-to-bottom fenestrations in cisternae, closed-headed arrows indicate budding structures at the nuclear envelope, and pink arrow shows frustrated budding/fusion intermediate. **(E ii, iii, iv, F i, iv, v, and G i, iv, v)** 3D rendering of segmentation. **(E–G i)** Scale bar, 1 µm.

---

for each KO. This indicates genotype is a key contributor to variance and highlights variation between secretomes in each line (Fig. S4 B). The two individual KO clones for each golgin were pooled using a linear regression model to increase statistical power. Log fold change in protein abundance was plotted against P value to identify proteins whose secretion is significantly altered in KO cultures compared with WT (Fig. 5 A).

In total, 482 proteins and 427 proteins showed altered raw abundance (P < 0.05) in the secretome of GMAP210 KO and Golgin-160 KO cultures, respectively, compared with WT cells (Fig. 5 A and Fig. S4 C). GO-term analysis revealed that the top 15 biological pathways affected in both golgin KO samples included collagen fibril and ECM organization, consistent with the ECM defects described above, as well as cell adhesion and migration (Table 1). Proteins involved in osteoblast and chondrocyte differentiation were also impacted by golgin loss. This was not surprising for the GMAP210 KOs given the skeletal phenotypes reported in animal models and human disease (Bird et al., 2018; Costantini et al., 2021; Del Pino et al., 2021; Follit et al., 2008; Medina et al., 2020; Qian et al., 2021; Smits et al., 2010; Upadhyai et al., 2021; Wehrle et al., 2019; Yamaguchi et al., 2021), but intriguingly, a similar effect was seen following the loss of Golgin-160, which has not been reported to cause a skeletal phenotype in mice (Banu et al., 2002; Bentson et al., 2013). Pathways associated with proteolysis and signaling appeared to be impacted by the loss of Golgin-160 specifically (Table 1).

Examination of the individual proteins most heavily affected by loss of each golgin shows a clear depletion of ECM components and regulators in the media of both KO cultures, with a high degree of overlap between GMAP210 and Golgin-160 KOs (Fig. 5 A—proteins impacted in the same way by each KO circled blue; Fig. S4 C). For example, the protein family showing the greatest depletion in both mutant medium fractions were the proteoglycans (Fig. 5 A, circled green). Biglycan (BGN), decorin, and versican all showed reduced abundance in both KO samples, with aggrecan also depleted in the Golgin-160 KO and testican (SPOCK1) impacted in the GMAP210 KOs. This suggests the proteoglycans are particularly susceptible to Golgi dysfunction. Collagen types IV, V, and VI, fibronectin, and nidogen-2 also showed reduced abundance in the media from both KO lines compared with WT (Fig. 4 A). Similarly, the secretion of matrix-regulating enzymes such as FAM20C and PLOD2, as well as signaling molecules like TGFb, growth differentiation factor 6, and LTBP2, was significantly downregulated in both KOs. Surprisingly, the protein showing the greatest increase in secretion in both KO lines was the intracellular protein Rho GDP Dissociation Inhibitor Beta (ARHGDIB), indicating aberrant translocation to the extracellular space.

In addition to these similarities, we also observed many golgin-specific changes in this data set (Fig. 5 A—differentially impacted proteins circled in purple; Fig. S4 C). For instance, thrombospondin secretion is more greatly affected in GMAP210 KO cultures, while Golgin-160 KO cells fail to efficiently secrete numerous additional ECM proteins and regulators like laminins and TIMP metalloproteases. Several components of the complement system also showed reduced secretion in the Golgin-160 KO cell lines, implying this pathway may be impaired following Golgin-160 loss (Fig. 5, A ii and Fig. S4 C). Overall, most proteins impacted by Golgin-160 loss showed reduced rather than increased abundance in culture media. In GMAP210 KO cells on the other hand, there were a significant number of proteins showing increased secretion, although these tended not to be ECM components. Of note, several lysosomal enzymes were released from GMAP210 KO cells suggesting either poor sorting at the Golgi or increased lysosome secretion (Fig. 5, A ii and Fig. S4 C). In conclusion, although the secretion of a large subset of ECM proteins is broadly susceptible to loss of golgin function, there are also cargoes that show specificity with respect to their reliance on particular golgins.

## Golgin KOs induce changes in gene expression for some secreted proteins

Changes in secreted protein abundance in cell culture medium could arise due to the altered gene expression of the cargoes or altered trafficking to the cell surface. To test the former, we performed RT-PCR to measure the expression of cargo proteins identified in the secretome analysis. The proteins showing the greatest negative and positive fold change in abundance in the media of the KO cells were BGN and ARHGDIB, respectively (Fig. 5 A). Consistent with this, we found that the expression of mRNA encoding each protein mirrored these changes, with BGN expression reduced and ARHGDIB expression increased in KO clones (Fig. 5, B and E). The expression of decorin mRNA, a protein closely related to BGN, was also reduced alongside secretion (Fig. 5 D). To determine whether gene expression correlated with secretion more widely, four more targets were tested by RT-PCR: THBS1, which is only affected in GMAP210 KO cells; PLOD1, which is only impacted by Golgin-160 loss; and SPARC and FAM20C, which show reduced secretion in both KO lines. The expression of FAM20C was reduced in all KO clones, in line with secretion (Fig. 5 F). However, SPARC and PLOD1 expression were not significantly altered in any of the KO lines (Fig. 5, C and G) and THBS1 expression was not consistently impacted (Fig. 5 H). Gene expression levels therefore do not fully predict secretion patterns for all targets, suggesting golgin loss can lead to changes to ECM secretion in different ways.

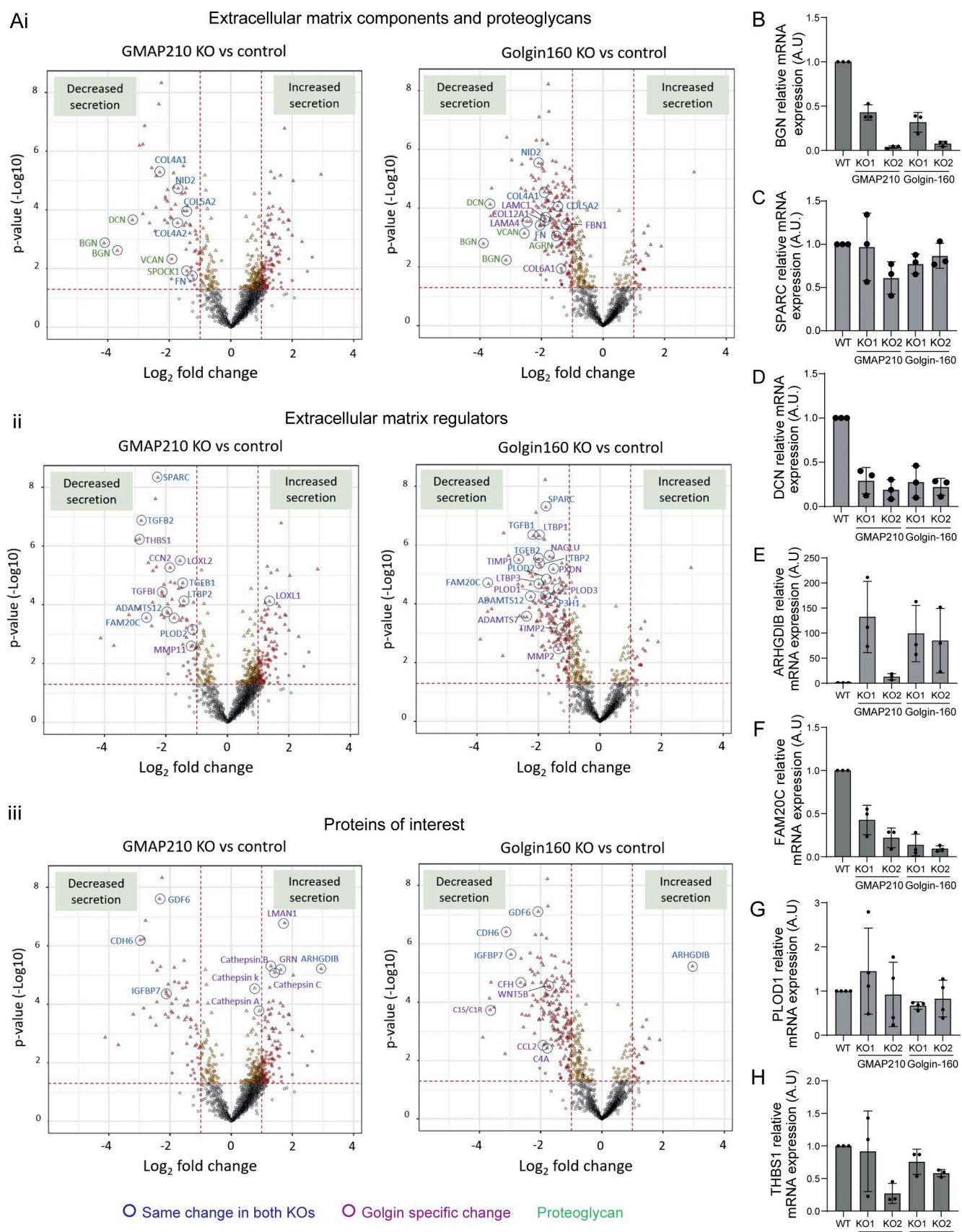

**Figure 5. General secretion is differentially impacted by loss of GMAP210 or Golgin-160. (A)** Volcano plots showing log-fold change in protein abundance in the media from KO versus WT cultures plotted against significance across three independent replicate experiments. The results from the two golgin KO clonal

cell lines were pooled to increase power. Lines and dot colors on the graph are for P < 0.05 and >1 or less than –1 LogFC. Triangular points show hits, which pass an FDR-corrected P < 0.05. **(i–iii)** Structural ECM proteins (i), ECM regulators (ii), and other proteins of interest (iii) have been highlighted. Proteins significantly changed in the same way in both KO cultures are labeled in dark blue, while proteins impacted in a golgin-specific way are labeled in purple. The proteoglycan family is indicated by green text. **(B–H)** RT-PCR analysis of the gene expression of (B) BGN, (C) SPARC, (D) DCN, (E) ARHGDIB, (F) FAM20C, (G) PLOD1, and (H) THBS1. Expression levels normalized to the expression of the housekeeping gene YWHAZ. Individual data points represent biological replicate experiments (*n* = 3–4). Bars show the mean and standard deviation.

## BGN trafficking is unaffected by golgin loss

We next wanted to determine the direct impact of our golgin mutations on cargo transport and modification. For this, we focused on BGN as the protein showing the greatest fold change in secretion in all lines. Stable cell lines were generated expressing a construct encoding BGN tagged with both mScarlet-i and a streptavidin-binding protein (SBP) to allow live imaging of exogenous BGN transport under the control of the RUSH trafficking system (Boncompain et al., 2021). In this system, tagged BGN is expressed alongside a streptavidin-tagged ER hook, which binds the SBP tag on the BGN and holds it in the ER. The addition of biotin prior to imaging is then used to outcompete this interaction and release BGN for anterograde trafficking. In these experiments, the ER hook (not fluorescently labeled) is transiently expressed on a bicistronic vector that also encodes BFP-tagged mannosidase II (ManII-BFP) to provide a Golgi marker.

In the absence of ER hook, stably expressed BGN-SBP-mSc was localized to both the ER and Golgi in all cell lines (Fig. S5). After transient transfection with the ER hook, successful anchoring of the tagged BGN was confirmed by localization of fluorescence to the ER at the steady state (Fig. 6, A i, B i, and C i; T 00:00). Upon the addition of biotin, BGN began to accumulate in cytoplasmic punctae, presumed to be ER exit sites, before moving toward the Golgi region in punctate carriers (Videos 4, 5, and 6; and Fig. 6, A ii, B ii, and C ii). Arrival at the Golgi region occurred ~14 min after biotin addition in all cases (Fig. 6 D). Here, the BGN signal initially accumulated adjacent to the ManII-BFP–labeled structures, but then showed increasing colocalization with ManII-BFP as it progressed through the stack (Videos 4, 5, and 6; Fig. 6, A i, B i, and C i; and Fig. S5 B). Finally, ~20 min after arrival at the Golgi, the emergence of post-Golgi carriers carrying BGN to the cell periphery was readily observed (Fig. 6, A iii, B iii, C iii, and E). Overall, transport kinetics were similar in all lines tested (Fig. 6, D and E), suggesting BGN secretion is unimpeded by loss of GMAP210 or Golgin-160, at least in an overexpression bulk transport system.

## SPARC trafficking is altered in the absence of GMAP210 and Golgin-160

As BGN expression levels were reduced in KO cells, we also tested a second cargo, SPARC, whose secretion was significantly reduced in KO cultures without any corresponding change in mRNA expression. Stable cell lines expressing SPARC-SBP-mSc were made, and live-imaging RUSH assays were performed as for BGN. Interestingly, this time we observed clear changes in SPARC transport dynamics upon golgin loss (Fig. 7; and Videos 7, 8, and 9).

In WT cells, upon biotin addition, SPARC appeared to accumulate in the Golgi without the appearance of discernible ER–

Golgi carriers (Video 7, Fig. 7 A, and Fig. S5 C). This is consistent with the "short-loop" pathway of ER–Golgi transport taken by collagen type I and described by McCaughey et al. (2019). In KO cells, however, SPARC accumulated in highly dynamic peripheral puncta just prior to, or concurrent with, its enrichment at the Golgi. Many of these puncta either moved toward and fused with the Golgi themselves or budded smaller ER–Golgi carriers. Other puncta persisted as stable structures throughout the movie and were ManII-BFP–negative, suggesting they were not Golgi elements. Despite transporting via an alternative route, the overall kinetics of ER–Golgi transport was the same between WT and KO cells (Fig. 7 D).

With respect to the Golgi exit, post-Golgi carriers could only be identified in 30–50% of KO cells during 45 min of imaging, in contrast to 80% of WT cells (Fig. 7, D iii). This implies Golgi transit is delayed or impaired in approximately half of cells, consistent with the proteomics results. Indeed, the study of the steady-state localization of SPARC-SBP-mSc in the absence of a RUSH hook showed that Golgi transit defects are so severe in Golgin-160 KO cells that it accumulates in the Golgi (Fig. 7 E). Altogether, our RUSH assays demonstrate that golgin loss impacts different cargoes in distinct ways.

## Post-translational modification of BGN is impaired in golgin KO cell lines

The interconnectivity of Golgi cisternae and their spatial organization is important to support and optimize the function of Golgi-resident enzymes that post-translationally modify cargoes (Zhang and Wang, 2016). ECM proteins are often highly modified, and these modifications are important to their assembly (Adams, 2023). We therefore sought to determine whether the disruption to Golgi organization in our mutants impacts ECM protein modification, which may in turn contribute to assembly defects. Proteoglycans are heavily modified by the addition of large, sulfated GAG chains, making them highly susceptible to Golgi dysfunction. We therefore continued to focus on BGN as a model cargo.

To look for changes in modification, cells stably expressing BGN-SBP-mSc were grown in serum-free medium overnight before collecting the media and lysate fractions of cultures for western blotting. Immunoblotting for both BGN and the mScarlet-i tag showed that exogenous BGN is secreted by all mutant lines (Fig. 8 A), consistent with the RUSH assays. Furthermore, the ratio of intracellular to extracellular protein was maintained, suggesting secretion rates are similar (Fig. 8 B). We did, however, notice a shift in the molecular weight distribution of secreted BGN in mutant cultures. In WT conditioned media, secreted BGN-SBP-mSc was identifiable as a broad band at around 250 kDa, with a small amount of protein running at 150

Table 1.  **GO-term analysis of golgin secretome data**

| Term | Count | Percent | P value | List total | Fold enrichment | Bonferroni | Benjamini | FDR |
|---|---|---|---|---|---|---|---|---|
| **Golgin-160 KO cell secretome GO-term analysis** | | | | | | | | |
| GO:0007155~cell adhesion | 41 | 16.4 | 3.80E-19 | 244 | 5.7676439124S5265 | 8.70E-16 | 8.70E-16 | 8.39E-16 |
| GO:0030199~collagen fibril organization | 17 | 6.8 | 1.52E-15 | 244 | 17.6098573557589 | 3.56E-12 | 1.75E-12 | 1.68E-12 |
| GO:0030335~positive regulation of cell migration | 20 | 8 | 1.16E-09 | 244 | 5.9747037514S821 | 2.66E-06 | 8.88E-07 | 8.55E-07 |
| GO:0030198~extracellular matrix organization | 17 | 6.8 | 2.24E-09 | 244 | 7.13662640207075 | 5.13E-06 | 1.05E-06 | 1.01E-06 |
| GO:0016477~cell migration | 20 | 8 | 2.29E-09 | 244 | 5.73829461021346 | 5.25E-06 | 1.05E-06 | 1.01E-06 |
| GO:0071711~basement membrane organization | 7 | 2.8 | 2.25E-07 | 244 | 25.37891207153S | 5.15E-04 | 8.59E-05 | 8.27E-05 |
| GO:0007275~multicellular organism development | 13 | 5.2 | 2.65E-07 | 244 | 7.2007627504S537 | 6.07E-04 | 8.67E-05 | 8.36E-05 |
| GO:0006508~proteolysis | 22 | 8.799999999999999 | 3.31E-07 | 244 | 3.8313766196S781 | 7.58E-04 | 9.48E-05 | 9.14E-05 |
| GO:0002062~chondrocyte differentiation | 9 | 3.5999999999999999 | 4.46E-07 | 244 | 12.8189402810304 | 0.00102162810602268 | 1.14E-04 | 1.09E-04 |
| GO:0007179~transforming growth factor beta receptor signaling pathway | 11 | 4.3999999999999999 | 6.53E-07 | 244 | 8.51830335826834 | 0.00149669542098818 | 1.50E-04 | 1.44E-04 |
| GO:0006024~glycosaminoglycan biosynthetic process | 8 | 3.2 | 1.12E-06 | 244 | 14.5022354694485 | 0.00256771715921932 | 2.34E-04 | 2.25E-04 |
| GO:0035987~endodermal cell differentiation | 7 | 2.8 | 2.02E-06 | 244 | 18.0108408249603 | 0.00461716250691446 | 3.86E-04 | 3.72E-04 |
| GO:0030509~BMP signaling pathway | 10 | 4 | 2.94E-06 | 244 | 8.39603310612597 | 0.0067130316586329 | 5.18E-04 | 4.99E-04 |
| GO:0051897~positive regulation of protein kinase B signaling | 13 | 5.2 | 3.80E-06 | 244 | 5.60491803278688 | 0.00866714018326186 | 6.22E-04 | 5.99E-04 |
| GO:0043406~positive regulation of MAP kinase activity | 9 | 3.5999999999999999 | 5.27E-06 | 244 | 9.3228656S893123 | 0.0120127981719709 | 8.06E-04 | 7.76E-04 |
| **GMAP210 KO cell secretome GO-term analysis** | | | | | | | | |
| GO:0061644~protein localization to CENP-A containing chromatin | 15 | 8.06451612903225 | 6.42E-26 | 182 | 89.1117216117216 | 1.07E-22 | 1.07E-22 | 1.05E-22 |
| GO:0006336~DNA replication-independent nucleosome assembly | 16 | 8.60215053763441 | 2.93E-24 | 182 | 63.368353683353 | 4.88E-21 | 1.63E-21 | 1.60E-21 |

Table 1. **GO-term analysis of golgin secretome data (*Continued*)**

| Term | Count | Percent | P value | List total | Fold enrichment | Bonferroni | Benjamini | FDR |
|------|-------|---------|---------|------------|-----------------|------------|-----------|-----|
| GO:0032200~telomere organization | 16 | 8.6021505376344 | 2.93E-24 | 182 | 63.368353683353 | 4.88E-21 | 1.63E-21 | 1.60E-21 |
| GO:0045653~negative regulation of megakaryocyte differentiation | 14 | 7.5268817204301 | 1.83E-22 | 182 | 74.853846153846 | 3.06E-19 | 7.65E-20 | 7.51E-20 |
| GO:0006335~DNA replication–dependent nucleosome assembly | 14 | 7.5268817204301 | 7.41E-19 | 182 | 46.783653846154 | 1.23E-15 | 2.47E-16 | 2.43E-16 |
| GO:0006352~DNA-templated transcription, initiation | 14 | 7.5268817204301 | 3.47E-17 | 182 | 36.514071294559 | 5.79E-14 | 9.65E-15 | 9.49E-15 |
| GO:0006334~nucleosome assembly | 21 | 11.290322580645 | 1.33E-16 | 182 | 13.132253711201 | 1.85E-13 | 3.18E-14 | 3.12E-14 |
| GO:0006325~chromatin organization | 21 | 11.290322580645 | 2.46E-11 | 182 | 6.9523696181233 | 4.10E-08 | 5.13E-09 | 5.04E-09 |
| GO:0030199~collagen fibril organization | 12 | 6.4516129032258 | 1.30E-10 | 182 | 16.665049236478 | 2.17E-07 | 2.41E-08 | 2.37E-08 |
| GO:0007155~cell adhesion | 25 | 13.440860215054 | 6.06E-10 | 182 | 4.7149059053820 | 1.01E-06 | 1.01E-07 | 9.93E-08 |
| GO:0030198~extracellular matrix organization | 12 | 6.4516129032258 | 1.71E-06 | 182 | 6.7537304800462 | 0.0028415775364992 | 2.59E-04 | 2.54E-04 |
| GO:0001649~osteoblast differentiation | 10 | 5.3763440860215 | 3.15E-06 | 182 | 8.3542239010989 | 0.0052298653635155 | 4.37E-04 | 4.29E-04 |
| GO:0001666~response to hypoxia | 11 | 5.9139784946237 | 6.35E-06 | 182 | 6.6456199168063 | 0.0105323325939843 | 8.14E-04 | 8.00E-04 |
| GO:0016477~cell migration | 13 | 6.9892473118280 | 1.19E-05 | 182 | 5.0005138746146 | 0.0196624232674336 | 0.0014184415926505 | 0.0013937656441281 |
| GO:0002062~chondrocyte differentiation | 7 | 3.7634408602150 | 1.31E-05 | 182 | 13.366758241758 | 0.0216774495503507 | 0.0014610475382515 | 0.0014356303039451476 |
| GO:0030335~positive regulation of cell migration | 12 | 6.4516129032258 | 4.21E-05 | 182 | 4.8060254352389 | 0.067714253052996 | 0.0043004981983049 | 0.0042256844923956 |

Proteins showing significantly altered abundance in KO cell conditioned media compared with WT conditioned media (P <0.01) were entered into the DAVID knowledgebase (https://davidbioinformatics.nih.gov/) and analyzed against the biological pathway GO-BP-DIRECT list for gene ontology-term enrichment. The 15 GO terms showing the most significant enrichment, as determined by the P value, are presented here.

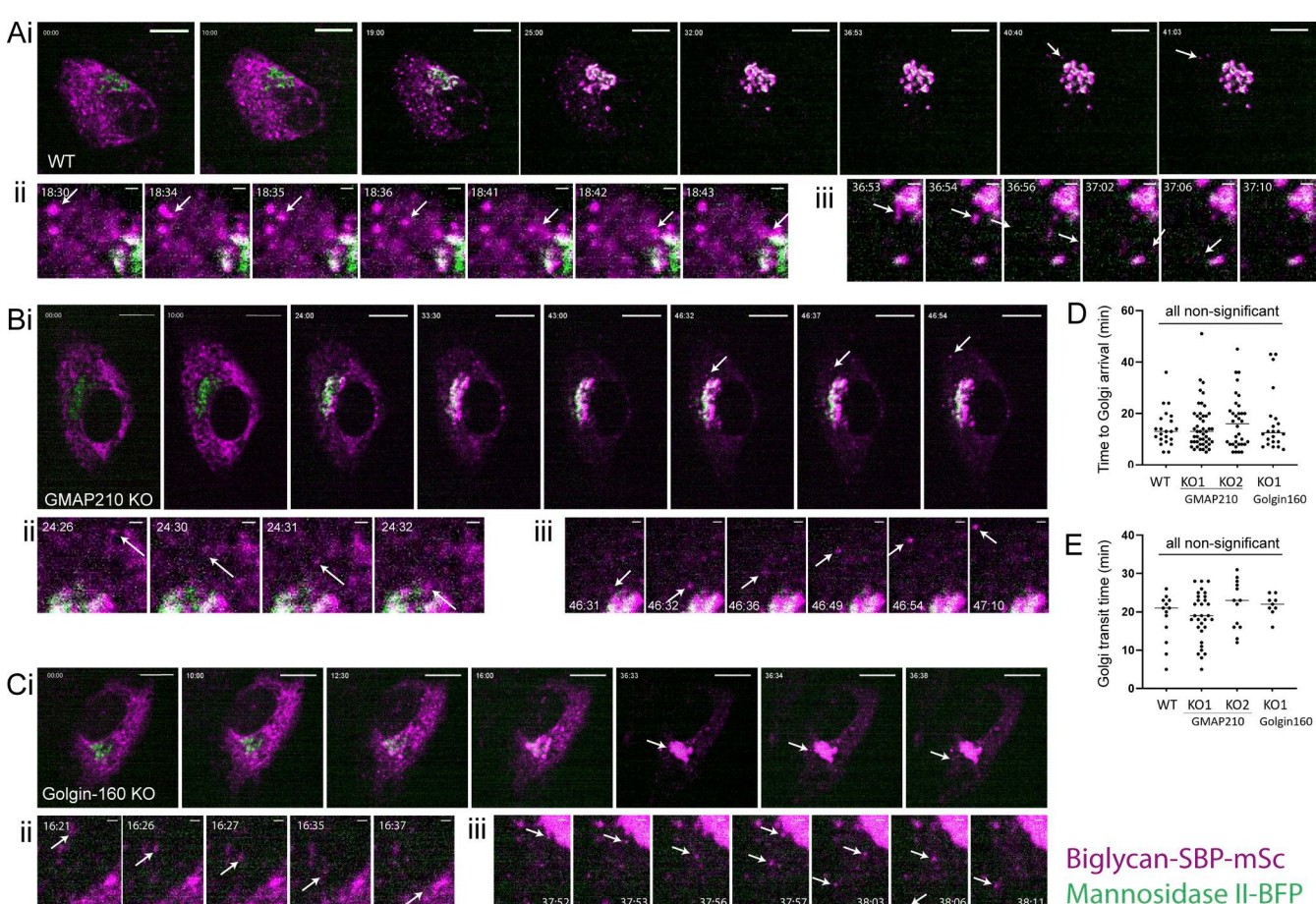

Figure 6. **BGN can traffic efficiently in golgin KO cells. (A–C)** BGN RUSH assay in WT (A), GMAP210 KO (B), and Golgin-160 KO (C) cells stably expressing BGN-SBP-mSc (magenta) and transiently transfected with an ER hook (not visible) and mannosidase II-BFP (green) prior to the experiment. Images are single-plane confocal images taken from time-lapse movies of BGN transport after release from the ER by biotin addition at T 00:00. Time after biotin addition is indicated in top left corner as mm:ss. **(A i, B i, and C i)** White arrows highlight the emergence of post-Golgi carriers. Scale bar, 10 µm. **(A ii, B ii, and C ii)** Crop showing incidence of ER–Golgi transport of BGN as highlighted by the arrow. Scale bar, 1 µm. **(A iii, B iii, and C iii)** Crop showing incidence of post-Golgi transport of BGN with carrier highlighted by the arrow. Scale bar, 1 µm. **(D)** Quantification of ER–Golgi transport, measured as time between biotin addition and the appearance of the BGN signal adjacent to mannosidase II-BFP label. **(E)** Quantification of Golgi transit time, measured as the time between BGN enrichment adjacent to mannosidase II-BFP signal and the emergence of a visible post-Golgi carrier. **(D–E)** All quantification performed on live movies as represented in A–C. Individual data points represent individual cells imaged across six independent experiments, and bars show the mean and standard deviation. **(D)** Data were subjected to a Shapiro–Wilk test for normality (failed) and then a nested one-way ANOVA with the Kruskal–Wallis test. **(E)** Data were subjected to a Shapiro–Wilk test for normality (passed) and then a nested one-way ANOVA with Dunnett's test for multiple comparisons to generate P values.

kDa. In all KO lines, however, a higher proportion of the protein ran at the lower molecular weight (Fig. 8 A). Calculation of the normalized ratio of 250- versus 150-kDa protein confirmed this observation (Fig. 8 C). Blotting with the higher affinity RFP antibody also showed the presence of fully unmodified BGN-SBP-mSc at 75 kDa in the medium fraction, which was again secreted to a greater extent by KO cells (Fig. 8 A). These data suggest BGN post-translational modification is perturbed in a similar way in the absence of either golgin.

To better understand the nature of the BGN modification defect, we first assessed N-glycosylation by subjecting secreted BGN-SBP-mSc to digestion by PNGase F, which cleaves N-glycan chains. After treatment, an identical small shift in BGN molecular weight was observed in all samples, indicating that this modification is intact in KO cultures (Fig. 8 D). On the other hand, treatment of immunoprecipitated BGN with chondroitinase

ABC, which digests GAG chains, did result in the loss of all higher and mid-molecular weight forms (Fig. 8 E). Interestingly, there was some resistance to digest in the GMAP210 KO clones that we attribute to altered modifications affecting enzyme–substrate affinity. Thus, we conclude that glycanation of BGN is defective in the golgin KO cells.

One Golgi organizational feature common to both GMAP210 and Golgin-160 KO lines is fragmentation of the Golgi ribbon. To test whether this would be sufficient to impair GAG chain synthesis, we fragmented the Golgi in our cell lines using the microtubule-depolymerizing agent nocodazole and repeated our secretion assays. Consistent with other studies (Harada et al., 2024), drug treatment impaired BGN modification in WT cells (Fig. 8 F). Surprisingly, however, we also found that nocodazole treatment exacerbated the GAG modification defects in mutant cells, suggesting the observed fragmentation in KO lines was not

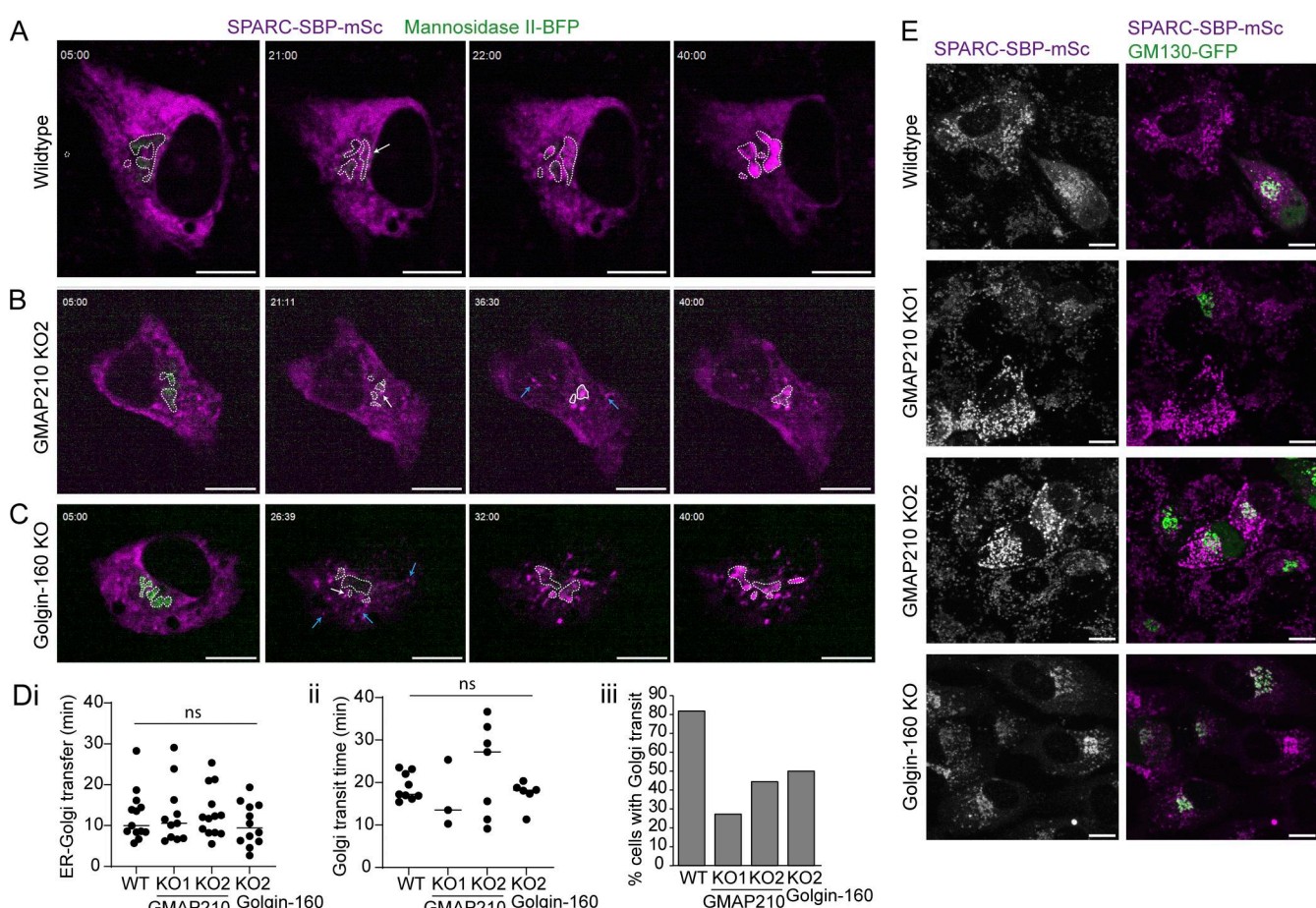

**Figure 7. SPARC traffics through alternative pathways in golgin KO cells. (A–D)** SPARC RUSH assay in WT (A), GMAP210 KO (B), and Golgin-160 KO (C) cells stably expressing SPARC-SBP-mSc (magenta) and transiently transfected with an ER hook (not visible) and mannosidase II-BFP (green, outlined by white dashed line) prior to the experiment. Images are single confocal planes taken from time-lapse movies of SPARC transport after release from the ER by biotin addition at T 00:00. Time after biotin addition is indicated in the top left corner as mm:ss. White arrows indicate SPARC accumulation adjacent to the mannosidase II–positive membranes. Blue arrows highlight peripheral puncta that initiate vesicular ER–Golgi transport. Scale bars, 10 μm. **(D i–iii)** Quantification of (i) time to arrival at Golgi after biotin addition, (ii) Golgi transit time, measured as the time between SPARC enrichment adjacent to mannosidase II-BFP signal and the emergence of a visible post-Golgi carrier, and (iii) the number of cells in which post-Golgi carriers were identified within 45 min of imaging time, from movies represented in A–C. Individual data points represent individual cells imaged across four independent experiments, and bars show the mean and standard deviation. **(i)** Data were subjected to a Shapiro–Wilk test for normality (failed) and then a nested one-way ANOVA with a Kruskal–Wallis test. **(ii)** Data were subjected to a Shapiro–Wilk test for normality (passed) and then a nested one-way ANOVA with Tukey's multiple comparison test. **(E)** Maximum projection confocal stacks of SPARC-SBP-mSc stable cell lines transfected with GM130-GFP to mark the Golgi. Scale bar, 10 μm.

fully pervasive. Overall, our studies show that GAG chain synthesis is impaired in the absence of both golgins tested.

## Discussion

In this study, we show that loss of two *cis*-Golgi–localized golgins causes distinct changes to Golgi morphology and function that ultimately lead to similar defects in ECM secretion, assembly, and organization. This is the first time that loss of Golgin-160 has been shown to impact ECM deposition and highlights the broad susceptibility of ECM cargoes to golgin loss. Altogether, our data suggest that the impact of golgin depletion on ECM assembly is manifold, with gene expression, protein secretion, protein modification, and ECM turnover all affected.

This study provides the most in-depth characterization of the Golgi structure in the absence of GMAP210 to date. In

agreement with previous reports, we observed that loss of GMAP210 results in compaction of Golgi structures in the perinuclear region, fragmentation of the Golgi ribbon, and cisternal dilation (Bird et al., 2018; Sato et al., 2015; Smits et al., 2010; Wehrle et al., 2019; Yamaguchi et al., 2021). We also note the concurrent unstacking of cisternae and build-up of tubulovesicular membranes in the peri-Golgi region. Altogether, this is indicative of a severe, general disruption to Golgi membrane dynamics. In 2014, Wong and Munro demonstrated that GMAP210 tethers GalNAcT2-containing vesicles and supports the correct localization of golgins giantin and GCC88 (Wong and Munro, 2014). Its loss may therefore impact enzyme sorting and transport directly, while indirectly triggering a cascade of problems in membrane trafficking by causing the mislocalization of other Golgi organizing proteins.

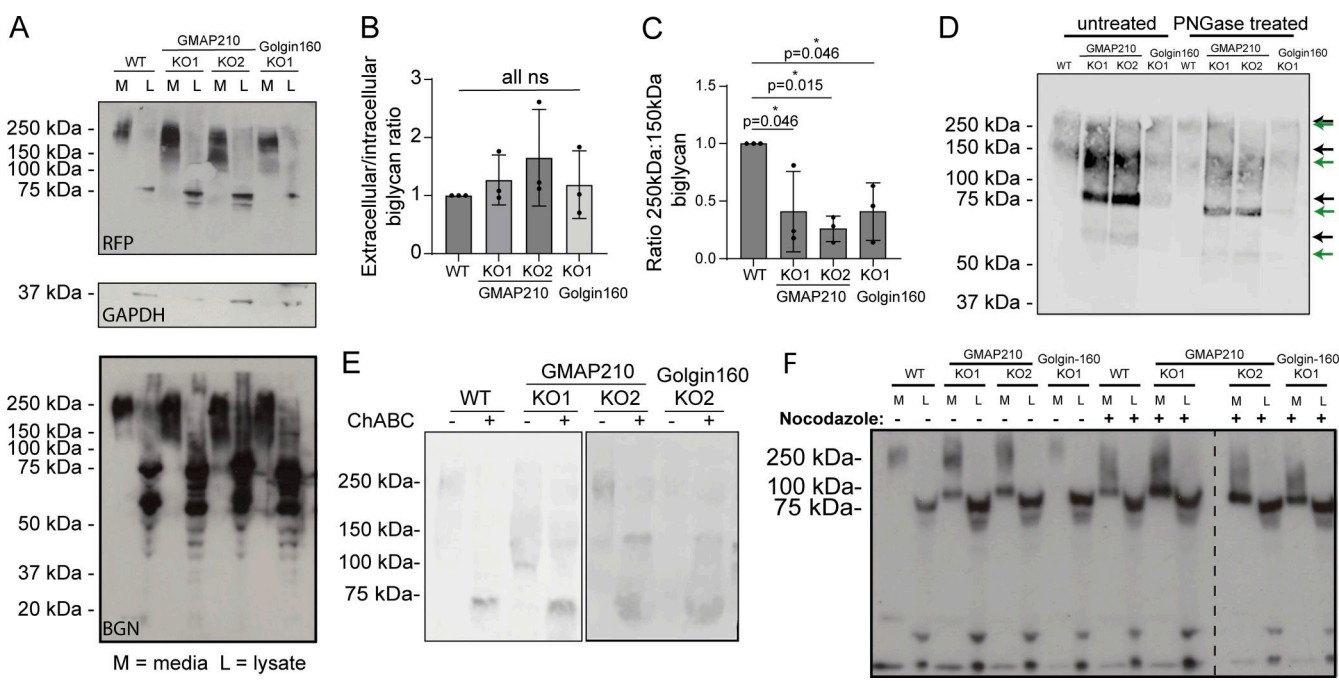

Figure 8. **BGN is undermodified in golgin KO cells. (A)** Western blots of medium (M) and lysate (L) samples taken from WT, GMAP210 KO, and Golgin-160 KO cells stably expressing BGN-SBP-mSc, probed with anti-RFP, GAPDH, and anti-BGN antibodies. Medium samples were collected over 16 h. **(B)** Quantification of the ratio between extracellular (M) and intracellular (L) BGN in WT and KO cultures, as determined by densitometry of the blots shown in A. **(C)** Quantification of the ratio between 250- and 150-kDa BGN bands present in the blots represented in A. **(B and C)** Dots represent measurements from independent experiments, and bars show the mean and SD. Data were subjected to a Shapiro–Wilk test for normality (passed) and then a nested one-way ANOVA with Dunnett's test for multiple comparisons to generate P values. **(D and E)** Western blots of medium and lysate samples from WT and KO lines stably expressing BGN-SBP-mSc. Medium samples were collected over 16 h and then treated with PNGase F (D) or chondroitinase ABC (E) prior to SDS-PAGE. Blots were probed with antibodies targeting the mScarlet-i tag (RFP antibody). **(D)** Arrows depict undigested (black) and PNGase F–digested (green) protein species. **(F)** Western blots of medium and lysate samples from WT and KO lines stably expressing BGN-SBP-mSc. Media were collected 5 h after the addition of DMSO (control) or nocodazole to cultures. Immunoblot for mScarlet-i tag (RFP antibody). Source data are available for this figure: SourceData F8.

Loss of Golgin-160 did not seem to affect Golgi stacking or cisternal flattening; however, there was a clear build-up of transport vesicles. Unfortunately, we were unable to confirm whether these were COPI vesicles at the light level as individual vesicles could not be distinguished in the peri-Golgi region. Regardless, the large number of vesicles in the peri-Golgi region in KO cells is consistent with a loss of vesicle tethering function, which may explain the large number of cargoes negatively impacted by Golgin-160 loss. A second feature of Golgi organization in Golgin-160 KO cells was fragmentation of the Golgi ribbon. It has been reported that Golgin-160 recruits dynein to Golgi membranes (Yadav et al., 2012). Microtubules are required to support lateral tethering of cisternae, and thus, reduced motor and microtubule recruitment in the absence of Golgin-160 may underlie this phenotype. Note that we did not observe the Golgi positioning or anterograde transport defects reported previously in another system (Yadav et al., 2012).

In humans, severe and hypomorphic mutations in the Trip11 gene encoding GMAP210 cause achondrogenesis type 1A (ACG1A) and odontochondrodysplasia (OCDC), respectively (Costantini et al., 2021; Del Pino et al., 2021; Medina et al., 2020; Qian et al., 2021; Upadhyai et al., 2021; Wehrle et al., 2019). These diseases are characterized by skeletal abnormalities such as shortened ribs and cranial–facial malformations. The study of Trip11 KO mouse models has ascertained that a primary driver of

pathology is defective ECM secretion during skeletal development (Bird et al., 2018; Follit et al., 2008; Smits et al., 2010; Yamaguchi et al., 2021). Here, we report a comparable attenuation of ECM secretion in our GMAP210 KO epithelial lines, with many of our phenotypes recapitulating those of in vivo and patient studies, including impaired collagen fibrillogenesis (Yamaguchi et al., 2021) and diminished production of highly glycanated proteoglycans (Wehrle et al., 2019). Interestingly, skeletal dysplasia has not been reported in existing studies of Golga3/Golgin-160 mutant mice (Banu et al., 2002; Bentson et al., 2013; Matsukuma et al., 1999), which seems incongruous with the similarities seen here between GMAP210 and Golgin-160 KO ECM defects. Compensatory mechanisms often mask severe phenotypes in golgin mutant animal models, which may explain this discrepancy (Bergen et al., 2017). Whether more subtle in vivo phenotypes, such as altered bone mass density, will be found with more extensive investigation remains to be determined.

Golgin-160 negatively impacted the secretion of a greater number of ECM components than GMAP210, including several laminins and metalloproteases. This implies it has a greater impact on transport. Consistent with this, Golgin-160 has been shown to bind to the Golgi-associated PDZ domain–containing protein PIST/GOPC (Hicks and Machamer, 2005), which supports transport of cargoes to the plasma membrane. More

specifically, the anterograde trafficking of the membrane-bound GLUT4 transporter (Williams et al., 2006) and β4 adrenergic receptor (Hicks et al., 2006) has been shown to require Golgin-160. Our results expand the list of Golgin-160–dependent cargoes in the context of soluble protein secretion and show that ECM components are particularly affected.

Overall, while Golgin-160 KO had a primarily negative impact on general secretion, more than half of the cargoes impacted by GMAP210 loss showed increased secretion in KO cells compared with WT. Interestingly, this was not the case for ECM proteins, which were almost always negatively impacted. One family of proteins specifically impacted by GMAP210 KO were the lysosomal enzymes cathepsins, four of which were aberrantly secreted by KO cells. This raises the possibility that the increased abundance of certain proteins in GMAP210 KO culture medium may stem from sorting defects. Interestingly, secreted cathepsins have been implicated in ECM degradation (Vidak et al., 2019; Vizovišek et al., 2019). Of note, two of those identified here (cathepsins B + K) have been shown to cleave targets such as SPARC (Podgorski et al., 2009), collagen type IV (Guinec et al., 1993), and fibronectin (Guinec et al., 1993), which were all reduced in abundance in the same KO media. Whether secreted cathepsins are contributing to the ECM phenotypes observed here therefore remains an intriguing open question, especially considering their potential as drug targets if found to contribute to disease pathology (Jangra et al., 2024).

Proteoglycans appear particularly susceptible to the loss of both GMAP210 and Golgin-160, with KO affecting both gene expression and glycanation of BGN. The biosynthetic pathway for GAG chains serves as an excellent example of the importance of Golgi organization (Hellicar et al., 2022; Ricard-Blum et al., 2024). Chains are built by enzymes organized sequentially across the Golgi stack, and competition between enzymes determines which types of GAG chains are built. Consequently, chain chemistry is determined by relative enzyme abundance, enzyme distribution, and trafficking speeds, making it particularly sensitive to a variety of Golgi perturbations (Adusumalli et al., 2021; Ahat et al., 2022; Chang et al., 2013). In our case, a common structural feature in both GMAP210 and Golgin-160 KO lines is Golgi fragmentation. Previously, this has been shown to disrupt the balance of GAG chain species (Ahat et al., 2022; Harada et al., 2024) and reduce overall GAG sulfation (Ahat et al., 2022). We also find that nocodazole-induced Golgi fragmentation in RPE1 cells impairs BGN GAG modification. Interestingly, nocodazole treatment of our mutant lines exacerbated glycan defects, suggesting that Golgi fragmentation is not fully pervasive. It is therefore worth considering the role of other golgin-specific changes to Golgi morphology.

The most prominent feature of Golgin-160 KO cells was the accumulation of transport vesicles. Loss of the COPI-binding COG complex has previously been shown to interfere with GAG chain length (Adusumalli et al., 2021), presumably due to impaired enzyme recycling. Thus, disruption to vesicular transport may contribute to glycan assembly defects in Golgin-160 KO cells. Meanwhile, a striking feature of GMAP210 KO cells was cisternal dilation, a feature also seen following the treatment of cells with actin-destabilizing drugs (Lázaro-Diéguez

et al., 2006). One such drug, cytochalasin B, also inhibits proteoglycan synthesis without impacting secretion (Lohmander et al., 1979). Thus, cisternal dilation may contribute to glycan undermodification in the KO cells, perhaps by impacting the frequency with which membrane-bound enzymes interact with soluble luminal cargo and so reducing the efficiency of modification. Overall, it seems likely that the combined effect of different perturbations to Golgi homeostasis in each mutant cell line contributes to impaired glycanation.

In conclusion, we propose that golgins have distinct, nonredundant functions but work collectively to create the correct physical and biochemical space to support efficient ECM protein secretion and modification.

## Materials and methods

### Cell culture
Telomerase immortalized human retinal pigment epithelial cells (hTERT-RPE1) were purchased from the American Type Culture Collection (ATCC) and grown in DMEM-F12-HAM (#11320-033; Life Technologies) supplemented with 10% decomplemented fetal bovine serum (FBS, A5256701; Gibco), passaging 1:10 every 3–4 days. HEK293T cells used for making lentivirus were grown in DMEM (D5796; Sigma-Aldrich) supplemented with 10% FBS. MC3T3-E1 cells (Subclone 4, CRL-2593; ATCC) were grown in MEMα (nucleosides, no ascorbate; catalog #A1049001; Life Technologies) supplemented with 10% decomplemented FBS (A5256701; Gibco). Transient transfections of RPE1 cells were performed using Lipofectamine 2000 (#11668027; Invitrogen) and 2 µg DNA per 35-mm dish according to the manufacturer's instructions. Stable cell lines were generated using the Lenti-X Packaging Single Shots kit (#631275; Takara) according to kit instructions and as described in the next section.

### Generating KO cell lines
#### Transfection with RNPs
To generate GMAP210 KO lines, the IDT ALT-R CRISPR system was used according to the manufacturer's protocols. Two crRNAs with sequences 5′-ATCCTGGAGTGCAATCTGTC-3′ (designed by IDT and targeting exon 4) and 5′-TATTTGGTCGGA TCGCCCGG-3′ (designed by Benchling targeting exon 1) were ordered from IDT. To duplex the gRNAs, 1 µl of 1 nmol/µl Alt-R CRISPR/Cas9 crRNA and 1 µl of 1 nmol/µl Alt-R CRISPR/Cas9 tracrRNA (#1072533; IDT) were mixed with 98 µl duplex buffer (#1072570; IDT) and heated to 95°C for 5 min before cooling on the bench. Alt-R S.p. Cas9 Nuclease V3 (#10010588; IDT LOT 000088243) was diluted to 1 µM in Cas9 working buffer (20 nM HEPES, 150 nM KCl, pH 7.5). To make the RNP, 1.5 µl duplexed gRNA + 1.5 µl 1 µM Cas9 + 0.6 µl Cas9 PLUS reagent (#CMAX00001; Invitrogen Lipofectamine CRISPRMAX) + 21.4 µl Opti-MEM were mixed per transfection. RNP mixes were left for 5 min at room temperature, and then, 25 µl RNP mix + 1.2 µl CRISPRMAX (#CMAX00001; Invitrogen Lipofectamine CRISP-RMAX) + 23.8 µl of Opti-MEM (#31985-070; Gibco) were added to a well of a 96-well plate. This was left to incubate for 20 min at room temperature. Meanwhile, RPE1 cells were trypsinized, counted, centrifuged at 1,000 × g for 3 min, and

then resuspended at a concentration of 20,000 cells per 100 µl. After the 20-min transfection incubation, 100 µl cell suspension (20,000 cells) was added to the wells and plates were put at 37°C, 5% CO$_2$. Cells were then expanded, and successful gene KO was determined by immunofluorescence staining of transfected cells.

MC3T3 cells were transfected as above using IDT predesigned gRNAs: *Trip11* gRNA sequence 5'-CCAGATTTCGAACTTCACGA-3' targeting exon 1, and *Golga3* gRNA sequence 5'-GGCCCGACT GCCTGATCAAC-3' targeting exon 2.

### Transfection with lentiCRISPR

Golgin-160 KO lines were generated using lentiCRISPRv2 constructs. To make the construct, two single-stranded oligos targeting exon 20 of the GOLGA3 gene (transcript GOLGA3-101 on ensemble) plus vector overhangs were ordered from IDT with sequences: F1: 5'-CACCGCTGGACTTGACGGAGCAGCA-3'; R1: 5'-AAACTGCTGCTCCGTCAAGTCCAGC-3'. Oligos were annealed in T4 ligation buffer (#B0202A; NEB) with T4 PNK (M0201S; NEB) for 30 min at 37°C followed by a 5-min incubation at 95°C and cooling by ramping down to 25°C at 5°C/min. Annealed oligos were ligated into the lentiCRISPRv2 vector (Plasmid #52961; Addgene) overnight using T4 ligation buffer (#B0202A; NEB) and T4 ligase (#M0202S; NEB). Ligations were transformed into One Shot Stbl3 Chemically Competent *E. coli* (#C737303; Invitrogen) according to the manufacturer's instructions, and bacterial cultures were grown on ampicillin-supplemented agar plates (#214530; BD). Successful ligation was determined by extracting constructs with a QIAprep Spin Miniprep kit (#27106; QIAGEN) from bacterial colonies, followed by sequencing (MWG Eurofins). The plasmid was then transfected into HEK293T cells using the Lenti-X Packaging Single Shots kit (#631278; Takara Bio) according to kit instructions. After 48 h, virus was harvested and 8 µg/ml polybrene (sc-134220; Santa Cruz Technology) added to a 1 ml aliquot. Media were aspirated from a 6-cm dish of RPE1 cells, and the virus aliquot was added to cells for incubation for 1 h at 37°C. DMEM-F12-HAM supplemented with 8 µg/ml polybrene was then added to cells, and cultures were incubated for 48 h at 37°C. Transfected cells were selected at the next passage by supplementing media with 10 µg/ml puromycin dihydrochloride (sc-1080701; Santa Cruz Technologies). Successful gene editing was determined by immunofluorescence staining of transfected cells.

### Making clones

To identify gene-edited cells, CRISPR/Cas9-transfected cell lines were single cell–sorted into a 96-well dish. Clones were expanded, and KO clones were confirmed by immunofluorescence and western blot. DNA was extracted from confirmed KO clones using a PureLink Genomic DNA Mini kit (#K1820-02; Invitrogen), and the CRISPR target region was amplified by PCR using Q5 Hot Start High-Fidelity 2× Master Mix (#M0493L; NEB) according to the manufacturer's instructions and touchdown PCR program: (1) 95°C for 3 min, (2) melting at 95°C for 25 s, (3) annealing at 72°C for 25 s, (4) extension at 72°C for 45 s, (5) repeating steps 2–4 reducing the annealing temperature by 0.5°C per cycle x 20 cycles, (6) melting at 95°C for 25 s, (7)

annealing at 72°C for 25 s, (8) extension at 72°C for 45 s, (9) repeating steps 6–8 ×25 cycles, (10) 72°C for 5 min, and (11) holding at 4°C.

Golgin-160/GOLGA3 genotyping primers: Fwd 5'-TGGGAG GTGGATCAGAAAGA-3', Rev 5'-ACCCTCCCATTTGCTGTAGT-3'; GMAP210/TRIP11 genotyping primers for exon 4: Fwd 5'-GTT GATAGAGGACCATCATTGGA-3', Rev 5'-TACACCAGCTCCTGA AGGTA-3'; GMAP210/TRIP11 primers for exon 1: Fwd 5'-TTC GTGGGGAATGAGCAGAAG-3', Rev 5'-CAAGAGGAGGTGTGTGAA GAAA-3'.

PCR products were purified using the QIAquick PCR purification kit (#28104; QIAGEN) and A-tailed by mixing 3 µl PCR product + 1 µl 10× ThermoPol buffer (M0267S; NEB) + 0.2 µl ATP + 1 µl Taq polymerase and incubating the reaction at 70°C for 30 min. This reaction was then ligated into a pGEM t-easy vector with T4 ligase and T4 ligation buffer (#A1360; Promega) so that individual alleles could be sequenced. Sequencing was performed by MWG Eurofins using their stock T7 forward primer.

### Cloning and constructs

To make BGN-SBP-mSc, two gene blocks were designed using HiFi NEBuilder to encode the required tagged BGN protein plus accompanying flanking sequences for sequential assembly into the pLVXNeo vector (#632181; Takara Bio) using the NEB HiFi assembly system. The tagged BGN coding region consisted of the BGN mRNA sequence (transcript ID: ENST00000331595.9, CCDS: CCDS14721) followed by linker 5'-GSAGSAAGSGEF-3' (Waldo et al., 1999), the SBP sequence, linker 5'-GGGGSGGGGS-3', and the mScarlet-i coding sequence. pLVXNeo was digested with XhoI (#R0146S; NEB) in NEB 3.1 10× buffer (#B7203S; NEB) for 15 min at 37°C, and then, BsmBI-V2 (#R0739S; NEB) was added for a second digest at 55°C for 20 min. Reactions were heated to 80°C for 20 min to inactivate enzymes. Gene block 1 was assembled into the digested vector with the HiFi assembly mix (#E2621; NEB), and then, the plasmid was transformed into NEB 5-α cells (C2987H), both according to the manufacturer's instructions. Colonies were grown up in LB plus ampicillin, and plasmids were extracted using a QIAprep Spin Miniprep kit (#27106; QIAGEN). Successful assembly of insert into vector was confirmed by test digest with BsrGI and BamHI and sequencing. This intermediate vector was then digested with XhoI and assembled with the second block and confirmed as before.

SPARC-SBP-mSc was designed and synthesized using the same strategy as BGN-SBP-mSc, but only one gene block and BsmBI-XbaI digestion were required. The tagged SPARC was made up of the SPARC mRNA sequence (transcript ID: ENST00000231061.9, CCDS: CCDS4318) followed by linker 5'-GGAAGTGCTGGCTCC GCTGCTGGTTCTGGCGAATTC-3', the SBP sequence, linker 5'-GAAGGAAAATCAAGTGGGTCTGGGTCAGAGAGCAAAAGC ACC-3', and the mScarlet-i coding sequence.

Design of StrKDEL-IRES-mannosidase II-mTagBFP2 is detailed in McCaughey et al. (2021) and available on Addgene (Plasmid #165460). Design of the pro-col1a1-SBP-GFP plasmid is detailed in McCaughey et al. (2019) and available on Addgene (Plasmid #110726). GM130-GFP was a gift from Professor Martin Lowe, School of Biological Sciences, Faculty of Biology, Medicine and Health, University of Manchester, Manchester, UK.

## Preparation of the cell-derived matrix

To collect the cell-derived matrix, cells were grown to confluence (either on coverslips or plastic) and their media supplemented with 50 µg/ml L-ascorbic acid-2-phosphate (A8960; Sigma-Aldrich) for a further 7 days. Cells were then washed in PBS, and the cell layer was removed by incubation with prewarmed extraction buffer (20 mM $NH_4OH$ [#221228; Sigma-Aldrich] and 0.5% Triton X-100) for 3 min at room temperature. The dish was then washed three times with $dH_2O$ and incubated with 10 µg/ml DNase I (Roche) diluted in reaction buffer (10 mM Tris-HCl, pH 7.6, 2.5 mM $MgCl_2$, and 0.5 mM $CaCl_2$) for 30 min at 37°C. After a further three washes in water, the cell-derived matrix was either fixed for staining or AFM imaging, or collected in sample buffer for assay (see the appropriate section).

Primary antibodies are given in Table 2.

## Immunofluorescence and fixed cell microscopy

For antibody labeling, cells grown on autoclaved coverslips (Menzel #1.5; Thermo Fisher Scientific) were fixed with 4% PFA (P/0840/53; Thermo Fisher Scientific) for 10 min followed by permeabilization for 10 min with 0.1% TX-100 (#X100; Sigma-Aldrich) at room temperature. For ER antibodies, cells were fixed with MeOH (#32213; Sigma-Aldrich) for 3 min at –20°C. Cells were then blocked with 3% BSA (#A9647; Sigma-Aldrich) for 30 min before incubation with primary antibody for 30 min, three washes with PBS, and incubation with Alexa Fluor–conjugated secondary antibody (Invitrogen). After a further three washes, cells were labeled with 4,6-diamidino-2-phenylindole (D1306; Life Technologies) for two min, washed in PBS, and mounted on glass slides in MOWIOL 4–88 (#475904; Calbiochem). For immunolabeling of ECM, cells were PFA-fixed and stained without permeabilization. Collagen and fibronectin fibrillar features were analyzed using the matrix analysis Fiji macro TWOMBLI (Wershof et al., 2021) (see AFM below).

Widefield images of fixed cells were taken using an Olympus IX70 microscope with 60× 1.42 NA oil-immersion lens, Exfo 120 metal halide illumination with excitation, dichroic, and emission filters (Semrock), and a Photometrics Coolsnap HQ2 CCD, controlled by Volocity 5.4.1 (PerkinElmer). Chromatic shifts in images were registration-corrected using TetraSpeck fluorescent beads (Thermo Fisher Scientific). Images were acquired as Δ0.2-µm z-stacks. Imaging was performed at room temperature.

Confocal microscopy was performed using one of two systems: (1) a Leica SP5II AOBS (Acousto-Optical Beam Splitter) confocal laser scanning microscope equipped with 50-mW 405-nm laser, 150-mW Argon laser, 20-mW solid state yellow laser, 20-mW red He/Ne diode laser, photomultiplier tube detectors, and a 63× HCX PL APO CS oil objective; or (2) a Leica SP8 AOBS (Acousto-Optical Beam Splitter) confocal laser scanning microscope equipped with 65-mW Argon laser, 20-mW DPSS yellow laser, 10-mW red He/Ne, and 50-mW 405-nm diode laser, hybrid detectors, and a 63× HC PL APO CS2 oil objective. Images were acquired at 1,024 × 1,024 x-y resolution, averaging three line scans per channel, using LAS X software. Image stacks were taken with Δz of 0.2 µm. Imaging was performed at room temperature.

Golgi area and fragmentation were measured from confocal images using an ImageJ macro with the following procedures:

subtract background (rolling ball size 20), apply median filter (radius determined in individual experiments), apply threshold (set for each experiment), make binary, analyze particles (size >0.1).

## Electron microscopy

Cells were grown to confluence in 35-mm dishes and then fixed *in situ* in 2.5% glutaraldehyde/0.1 M cacodylate buffer for 20 min. Cells were then washed with 0.1 M cacodylate buffer for 10 min and fixed in 1% $OsO_4$, 1.5% potassium ferrocyanide, 0.1 M cacodylate buffer for 60 min. Cells were washed with 0.1 M cacodylate buffer twice, and with water twice for 10 min each. To negative stain, cells were incubated with 3% uranyl acetate for 20 min and washed again with water. Samples were then dehydrated by successive 10-min incubations with 70, 80, 90, 96, and 100% EtOH. After a second 100% EtOH incubation, all EtOH was removed and approximately 1 ml of a 50:50 mix of propylene oxide and epon was added to the dish. Samples were placed on a rocker for 2 h. This was exchanged with 100% epon before placing resin stubs in the dish and baking the samples at 70°C for 2 days. Stubs were extracted from the dish and trimmed. Ultrathin sections of 70 nm were cut using a diamond knife and a UC6 ultramicrotome (Leica Microsystems) and incubated with 10-nm gold nanoparticles to act as fiducials (#752584; Sigma-Aldrich). Sections were imaged using a Tecnai20 LaB6 200k kV twin lens transmission electron microscope (FEI) to collect a tilt series between –70° and +70° using a dedicated Fischione tomography holder and FEI software. Tomograms were reconstructed using IMOD© software. Alignment was computed using fiducial tracking, and tomograms were generated using 10 iterations with SIRT-like filter reconstruction. Segmentation was performed using Amira 2019.3 software.

## AFM

Cells were grown on coverslips, and the cell-derived matrix was processed as above and then fixed in 4% PFA (P/0840/53; Thermo Fisher Scientific) for 10 min at room temperature. Coverslips were rinsed with deionized water to remove salt crystals visible in the HS-AFM images and mounted onto SEM stubs. Excess water was blown off with compressed gas, and samples were stored at room temperature prior to HS-AFM imaging. Imaging was performed with a HS-AFM (Bristol Nanodynamics, custom-built), and a MSNL-10 cantilever chip (Bruker) was run using Bristol Nanodynamics microscope readback software version 11.02. 100 images in 10 × 10 raster scans were collected in three different areas of each coverslip. Imaging and analysis of the samples were performed blind. Fibrillar features were analyzed from the images of one raster scan from each biological replicate ($n$ = 3) with the matrix analysis Fiji macro TWOMBLI (Wershof et al., 2021). TWOMBLI combines the ridge detection and AnaMorf plug-ins to identify filamentous structures in images of ECM (Wershof et al., 2021). Masks are generated from identified filaments, and various parameters are measured from these. Curvature is measured as the mean change in angle moving along a fiber, within a defined window. Alignment is computed by the plug-in OrientationJ, which calculates the proportion of fibers within an image orientated in a

Table 2. **Primary antibodies**

| Target | Company | Catalog # | Lot # | Immunogen | Dilution | Fix |
|---|---|---|---|---|---|---|
| Golgin-160 C-term RRID: AB_10681049 | Abcam | ab96080 | GR323647 | Human GOLGA3 aa 1400 to C terminus | IF 1:1,000 WB 1:500 | PFA/MeOH |
| Golgin-160 N-term RRID:AB_10733640 | Proteintech | 21193-1-AP | 00014538 | 1–350 aa encoded by BC142658 | IF 1:1,000 WB 1:500 | PFA/MeOH |
| GMAP210 C-term RRID: AB_2686295 | Sigma-Aldrich Prestige | HPA070684 | R98121 | MLDDVQKKLMSLANSSEGKVDKVLMR NLFIGHFHTPKNQRHEVLRLMGSILGVRREEMEQLFHDDQ GSVTRWMTGWLGGG SKSVPNTPLRPNQQSV | IF 1:500 | PFA/MeOH |
| GMAP210 N-term RRID: AB_1078995 | Sigma-Aldrich Prestige | HPA002570 | A95744 | QSLGQVGGSLASLTGQISNFTKDMLMEGTEEVEAELPDSRTK EIEAIHAILRSENERLKKLCTDLEEKHEASEIQIKQQS TSYRNQLQQKEVEISHLKARQIALQDQLLKLQSAAQSVPS GAGVPATTASSSFAY | IF 1:500 | PFA/MeOH |
| GMAP210 | BD Bioscience | BD611712 | 47539 | Full-length protein | IF 1:1,000 | PFA/MeOH |
| TGN46 RRID: AB_324049 | Bio-Rad | AHP500 | 170720 | Full-length human protein | IF 1:1,000 | PFA/MeOH |
| Giantin RRID:AB_291560 | Covance | PRB-114C | Clone 19243. B23348 | N terminus, residues 1–469, of human giantin | IF 1:2,000 | PFA/MeOH |
| GM130 RRID:AB_398142 | BD Bioscience | BD610823 | 3161628 | Rat GM130 aa. 869–982 | IF 1:1,000 | PFA/MeOH |
| BGN | Abcam | ab188508 | Gr2a5102.3 | Proprietary information | WB 1:500 | - |
| RFP (mScarlet) RRID: AB_2631395 | ChromoTek | 6G6-100 | 51020014AB | Full-length protein | WB 1:1,000 | - |
| bCOP | Sigma-Aldrich | G6160 | Ascites | Synthetic peptide D1 of β-COP (aa. 701–715) | IF 1:200 | MeOH |
| LMAN1/ERGIC53 RRID: AB_477023 | Invitrogen | MA5-25345 | WH3358331 | Full-length human protein | IF 1:1,000 | MeOH |
| Fibronectin RRID: AB_476988 | Sigma-Aldrich | F7387 | Unknown | Fibronectin from human plasma | IF 1:1,000 | - |
| Collagen type 1 RRID: AB_10000511 | Novus Biologicals | NB600-408 | 38825 | Collagen I from human and bovine placenta | IF 1:1,000 WB 1:500 | PFA |
| A-Tubulin RRID: AB_477579 | Sigma-Aldrich | T5168 | Unknown | Sarkosyl-resistant filaments from *Strongylocentrotus purpuratus* (sea urchin) sperm axoneme | WB 1:2,000 | - |

similar direction. Identical TWOMBLI parameters were used for all samples. Periodicity was determined by extracting the peak profiles along fibrillar structures using Gwyddion 2.60. The period was calculated using Excel, and data were displayed/analyzed with GraphPad Prism 10.0.2.

### Live imaging of RUSH assays

For live-imaging RUSH experiments, cells stably expressing BGN-SBP-mSc or SPARC-SBP-mSc were grown to confluence in 35-mm MatTek glass-bottom imaging dishes (P35G-1.5-20C; MatTek) and transiently transfected with ManII-BFP_IRES_KDEL-streptavidin (McCaughey et al., 2021) 24 h prior to imaging. To image, growth media were replaced with 1 ml prewarmed FluoroBrite DMEM (A1896701; Thermo Fisher Scientific) and dishes were mounted on an Olympus IXplore spinning disk system in a light-excluding environmental control chamber (PECON) at 37°C. At T0, 1 ml FluoroBrite media containing 80 µM biotin was added to the dish (final concentration 40 µm) and imaging was immediately started, taking 120 frames every 10 s followed by 4,800 frames every 500 ms. Data were acquired using a 60× oil-immersion lens with 1.5 numerical aperture, Hamamatsu Fusion BT sCMOS cameras, diode lasers, TruFocus drift compensation, and Olympus CellSens imaging software. The frame at which mScarlet-i signal could first be seen accumulating adjacent to the BFP signal was used to measure transit to Golgi time. The frame in which a tubular carrier could first be seen emerging from the BFP signal (that then continues to the periphery) was used to calculate Golgi transit time.

### Secretion assays

To perform secretion assays, cells were seeded in 6-well dishes and grown to confluence before aspirating their growth medium and replacing this with 1 ml serum-free DMEM-F12-HAM. After overnight incubation at 37°C/5% $CO_2$, cells were put on ice, the media were collected into Eppendorf tubes, and the cell layer was washed with ice-cold PBS. Cells were then lysed in RIPA buffer (50 mM Tris-HCl, pH 7.5, 300 mM NaCl, 2% Triton X-100, 1% deoxycholate, 0.1% SDS, 1 mM EDTA) containing protease inhibitors (#539137; Millipore) on ice for 15 min on a rocker. Medium samples were centrifuged at 2,000 × $g$ for 2 min at 4°C

to remove dead cells, and the supernatant was collected. RIPA lysates were scraped up into Eppendorf tubes and centrifuged for 10 min at 13,000 × g/4°C, and the supernatant was collected. Media and lysates were then mixed with 4× Bolt LDS sample buffer (#B0007; Invitrogen), boiled for 10 min at 70°C, and run on SDS-PAGE gels as below.

For nocodazole secretion assays, cell media were replaced with 1 ml serum-free DMEM-F12-HAM supplemented with 20 μM nocodazole or an equivalent volume of DMSO as a vehicle control for 5 h. Beyond this point, cells were beginning to round and longer collections were not possible. Media and lysates were then collected as normal.

For normal cell lysate collection and western blotting, cells were lysed in RIPA buffer and processed as above. For cell-derived matrix samples, prewarmed sample buffer with reducing agent was added to the dish after cell extraction (above), the dishes were scraped, and the buffer was collected and boiled for 10 min at 70°C.

### Deglycosylation assays
WT and KO cells stably expressing BGN-SBP-mSc were grown to confluence in a 150-mm dish and media exchanged for serum-free media 16 h prior to the experiment. Media were then collected into Falcon tubes on ice and spun at 1,200 rpm for 2 min to pellet out cells. The supernatant was collected and supplemented with a protease inhibitor cocktail (#539137; Millipore), and BGN-SBP-mSc was extracted by RFP-trap. To trap, a 60 μl aliquot of ChromoTek RFP-trap Agarose beads (#rta; ChromoTek) was washed three times in a Falcon tube with dilution buffer (10 mM Tris-HCl, pH 7.4, 50 mM NaCl, 0.5 mM EDTA), centrifuging for 2 min at 2,000 rpm at 4°C each time to pellet the beads for buffer exchange. On the final wash, all buffer was removed and the medium supernatant added to the beads. Beads and sample were mixed on a rotator at 4°C for 2 h. Tubes were spun at 2,700 × g for 2 min at 4°C to pellet the beads and the media removed. Beads were then washed three times as before in dilution buffer supplemented with protease inhibitors. After the removal of the final wash, the beads were purged.

*For digest with PNGase (kit, NEB P0704S)*, 10× glycoprotein denaturation buffer was diluted to 1× with water, and then, 80 μl of this was added to the bead pellet. Beads and buffer were transferred to an Eppendorf tube and then heated at 100°C for 10 min. Samples were then put on ice for 10 s before centrifuging them at 2,700 × g for 2 min. A 60 μl aliquot of the supernatant was transferred to a new tube to act as an undigested control. The remaining 20 μl was transferred to a new tube, and contents were digested by adding 4 μl 2× glycobuffer 2, 4 μl 10% NP-40, 10 μl of water, and 2 μl of PNGase F and incubating the mix at 37°C for one h. Samples were cooled to room temperature, mixed with 4× Bolt LDS sample buffer (#B0007; Invitrogen), and analyzed by SDS-PAGE as below.

*For digest with chondroitinase ABC (Merck Life Science Limited)*, bead pellets were resuspended in 200 μl dilution buffer (50 mM Tris-HCl, pH 8, 60 mM sodium acetate, 0.02% BSA) and heated at 65°C for 30 min to heat-inactivate any residual proteases. Of this suspension, 100 μl was then taken and 0.3U chondroitinase ABC added (digested sample). The remaining 100 μl was left

untreated (undigested). All samples were incubated at 37°C overnight. 4× Bolt LDS sample buffer (#B0007; Invitrogen) was added, and samples were boiled at 70°C ready for western blotting as below.

### Western blotting
Samples were run in Bolt 4–12% Bis-Tris gels with MOPS running buffer for 40 min at 130 V. They were then transferred to 0.2-μM nitrocellulose membranes and blocked with 5% milk/TBST. Primary antibody incubations were run overnight at 4°C, and then, membranes were washed with TBS/0.05% Tween (Sigma-Aldrich) and incubated with HRP (Jackson Immuno-Research)- or fluorophore-conjugated (Invitrogen) secondary antibodies for 2 h prior to imaging by enhanced chemiluminescence (Promega ECL) and autoradiography films (GE Healthcare) or by fluorescence imaging (LI-COR Odyssey), respectively. For collagen blotting, samples were run on NuPAGE 3–8% Tris-acetate gels at 120 V for 1 h and transferred onto 0.45-μm polyvinylidene fluoride (PVDF) membranes overnight at 12 V and 4°C. Membrane total protein staining was performed with a total protein stain kit (LI-COR). Quantification of band intensity was performed with ImageJ or Empiria Studio 3.0.

### Matrisome proteomics
Cells were grown in 15-cm dishes for 1 wk in the presence of 50 μg/ml L-ascorbic acid-2-phosphate (A8960; Sigma-Aldrich). Cells were then extracted, and the cell-derived matrix was collected by adding 1 ml PBS to the plate and scraping for 30 s. Samples were transferred to tubes and frozen at –20°C prior to processing.

Lysis buffer (10% sodium dodecyl sulfate [Sigma-Aldrich] in 50 mM TEAB [Sigma-Aldrich] supplemented with protease and phosphatase inhibitors) was added (1:1 vol/vol). Samples were sonicated using a Covaris LE220+ sonicator using a 40 W lysis program with 100 cycles per burst, a duty factor of 40%, and a peak incident power of 500 W. Next, 50 μg of protein was reduced and alkylated using DTT (final concentration 5 mM) and IAA (final concentration 15 mM), respectively. Samples were acidified with $H_3PO_4$ (1.2% final concentration), then 7 volumes of binding buffer (90% methanol, 10% distilled water, 100 mM TEAB) was added, and the samples were loaded onto S-trap columns (ProtiFi) according to the manufacturer's protocol.

Column-bound proteins were washed in binding buffer and then digested with 5 μg of 0.8 μg/μl trypsin solution (Promega) diluted in digestion buffer (50 mM TEAB at pH 8.5). Peptides were eluted from the column in 65 μl digestion buffer, 65 μl 0.1% formic acid (FA) in distilled water, and finally in 30 μl 0.1% FA, 30% acetonitrile (ACN, Sigma-Aldrich). Peptides were desalted using oligo R3 resin beads (Thermo Fisher Scientific), according to the manufacturer's protocol, in a 96-well, 0.2-μm PVDF filter plate (Corning). The immobilized peptides were washed twice with 0.1% FA prior to 2 × 50 μl elutions in 0.1% FA with 30% ACN, and lyophilization was performed using a SpeedVac (Heto Cooling System).

Peptides were resuspended in 10 μl 0.1% FA in 5% ACN. Liquid chromatography (LC) separation was performed on a Thermo RSLC system consisting of an NCP3200RS nano pump,

WPS3000TPS autosampler, and TCC3000RS column oven configured with buffer A as 0.1% FA in water and buffer B as 0.1% DA in ACN. The analytical column (Waters nanoEase M/Z Peptide CSH C18 Column, 130 Å, 1.7 μm, 75 μm × 250 mm) was kept at 35°C and at a flow rate of 300 nl/min for 8 min. The injection valve was set to load before a separation consisting of a 105-min multistage gradient ranging from 2 to 65% of buffer B. The LC system was coupled to a Thermo Exploris 480 mass spectrometry system via a Thermo Nanospray Flex ion source. The nanospray voltage was set at 1,900 V, and the ion transfer tube temperature was set to 275°C. Data were acquired in a data-dependent manner using a fixed cycle time of 2 s, an expected peak width of 15 s, and a default charge state of +2. Full mass spectra were acquired in positive mode over a scan range of 300 to 1,750 Th, with a resolution of 120,000, a normalized automatic gain control (AGC) target of 300%, and a max fill time of 25 ms for a single microscan. Fragmentation data were obtained from signals with a charge state of +2 or +3 with an intensity over 5,000. They were then dynamically excluded from further analysis for a period of 15 s after a single acquisition within a 10-ppm window. Fragmentation spectra were acquired with a resolution of 15,000, a normalized collision energy of 30%, an AGC target of 300%, a first mass of 110 Th, and a maximum fill time of 25 ms for a single microscan. All data were collected in profile mode.

Mass spectrometry data were analyzed using MaxQuant (v1.6.14) using default parameters. All searches included the fixed modification for carbamidomethylation on cysteine residues resulting from IAA treatment. The variable modifications included in the search were oxidized methionine (monoisotopic mass change, +15.955 Da), hydroxyproline (15.995 Da), acetylation of the protein N terminus (42.011 Da), and phosphorylation of threonine, serine, and tyrosine (79.966 Da). A maximum of 2 missed cleavages per peptide were allowed. Peptides were searched against the UniProt database with a maximum false discovery rate (FDR) of 1%. Proteins were required to have a minimum FDR of 1% and at least 2 unique peptides in order to be accepted; known contaminants were also removed. Missing values were assumed to be due to low abundance. Abundance was compared using MSqRob where label-free quantification data were normalized using median peptide intensity (Goeminne et al., 2020). MSqRob was used using R (version 4.1.2). The remaining statistics were done using Prism (GraphPad Software, version 9.1.2).

### Secretome proteomics

Cells were seeded into 6-well dishes and grown to confluence. Cells were rinsed once with serum-free medium, taking care to aspirate as much media as possible each time. Exactly 1 ml serum-free F12 HAM was then added to cells overnight. After 24 h, cells were put on ice and 900 μl medium was removed from well and added to an Eppendorf tube on ice. Media were spun at 2,000 rpm for 5 min, and then, 800 μl was removed and put in a fresh tube on ice. To test protein concentration, 25 μl medium was used in a BSA assay prior to snap freezing and storage at –80°C.

For TMT labeling and high-pH reversed-phase chromatography, an equal volume (95 μl) of each sample was digested with trypsin (1.25 μg trypsin; 37°C, overnight), labeled with tandem mass tag (TMTpro) 16-plex reagents according to the manufacturer's protocol (LE11 5RG; Thermo Fisher Scientific), and the labeled samples were pooled. The pooled samples were desalted using a SepPak cartridge according to the manufacturer's instructions (Waters). The eluate from the SepPak cartridge was evaporated to dryness and resuspended in buffer A (20 mM ammonium hydroxide, pH 10) prior to fractionation by high-pH reversed-phase chromatography using an Ultimate 3000 LC system (Thermo Fisher Scientific). In brief, the sample was loaded onto an XBridge BEH C18 column (130Å, 3.5 μm, 2.1 × 150 mm, Waters) in buffer A and peptides were eluted with an increasing gradient of buffer B (20 mM ammonium hydroxide in ACN, pH 10) from 0 to 95% over 60 min. The resulting fractions (concatenated into 15 in total) were evaporated to dryness and resuspended in 1% FA prior to analysis by nano-LC MS/MS using an Orbitrap Fusion Lumos mass spectrometer (Thermo Fisher Scientific).

High-pH RP fractions were further fractionated using an Ultimate 3000 nano-LC system in line with an Orbitrap Fusion Lumos mass spectrometer (Thermo Fisher Scientific). In brief, peptides in 1% (vol/vol) FA were injected onto an Acclaim Pep-Map C18 nanotrap column (Thermo Fisher Scientific). After washing with 0.5% (vol/vol) ACN /0.1% (vol/vol) FA, peptides were resolved on a 250 mm × 75 μm Acclaim PepMap C18 reverse-phase analytical column (Thermo Fisher Scientific) over a 150-min organic gradient, using 7 gradient segments (1–6% solvent B over 1 min, 6–15% B over 5 8 min, 15–32% B over 58 min, 32–40% B over 5 min, 40–90% B over 1 min, held at 90% B for 6 min, and then reduced to 1% B over 1 min.) with a flow rate of 300 nl min$^{-1}$. Solvent A was 0.1% FA, and solvent B was aqueous 80% ACN in 0.1% FA. Peptides were ionized by nanoelectrospray ionization at 2.0 kV using a stainless-steel emitter with an internal diameter of 30 μm (Thermo Fisher Scientific) and a capillary temperature of 300°C.

All spectra were acquired using an Orbitrap Fusion Lumos mass spectrometer controlled by Xcalibur 3.0 software (Thermo Fisher Scientific) and operated in data-dependent acquisition mode using an SPS-MS3 workflow. FTMS1 spectra were collected at a resolution of 120,000, with an AGC target of 200,000 and a max injection time of 50 ms. Precursors were filtered with an intensity threshold of 5,000, according to charge state (to include charge states 2–7), and with monoisotopic peak determination set to Peptide. Previously interrogated precursors were excluded using a dynamic window (60 s ±10 ppm). The MS2 precursors were isolated with a quadrupole isolation window of 0.7 m/z. ITMS2 spectra were collected with an AGC target of 10,000, max injection time of 70 ms, and CID collision energy of 35%.

For FTMS3 analysis, the Orbitrap was operated at 50,000 resolution with an AGC target of 50,000 and a max injection time of 105 ms. Precursors were fragmented by high-energy collision dissociation at a normalized collision energy of 60% to ensure maximal TMT reporter ion yield. Synchronous precursor selection was enabled to include up to 10 MS2 fragment ions in the FTMS3 scan.

The raw data files were processed and quantified using Proteome Discoverer software v2.1 (Thermo Fisher Scientific) and searched against the UniProt Human database (downloaded

January 2023: 81579 entries) and the UniProt Bos taurus database (downloaded February 2023: 37505 entries) using the SEQUEST HT algorithm. Peptide precursor mass tolerance was set at 10 ppm, and MS/MS tolerance was set at 0.6 Da. Search criteria included oxidation of methionine (+15.995 Da), acetylation of the protein N terminus (+42.011 Da), and methionine loss plus acetylation of the protein N terminus (−89.03 Da) as variable modifications and carbamidomethylation of cysteine (+57.021 Da) and the addition of the TMTpro mass tag (+304.207) to peptide N termini and lysine as fixed modifications. Searches were performed with full tryptic digestion, and a maximum of two missed cleavages were allowed. The reverse database search option was enabled, and all data were filtered to satisfy FDR of 5%.

As samples are from the medium fraction of cultures, equal volumes of sample were analyzed rather than equal quantities of protein, and raw abundance was used without normalization. Statistical analysis was performed in R version 4.3.0. PCAs were calculated using the FactoMineR package, PCAs, and Sum abundance, and volcano plots were plotted in ggplot2. A linear mixed-effects model was fitted for each protein using the package lme4, with replicate number included as a random effect, and both clones included in a single variable for the KO condition to increase statistical power. P values were then FDR-adjusted using the Benjamini–Hochberg method to account for multiple testing. Note that this was used as a gold-standard metric for individual proteins rather than a threshold for statistical analysis, as it is understood to be an overly stringent method for most proteomics experiments (Pascovici et al., 2016). Downstream analysis was performed using QIAGEN IPA (QIAGEN, Inc., https://digitalinsights.qiagen.com/IPA), using P < 0.05 as a cutoff and the unfiltered user data as the reference set (Krämer et al., 2014). GO-term functional enrichment analysis was performed using the Database for Annotation, Visualization, and Integrated Discovery (https://www.david.niaid.nih.gov).

### qPCR

Cells were grown to confluence in 35-mm dishes, and then, RNA was extracted using RNeasy Plus Mini Kit (#74136; QIAGEN). Reverse transcription was performed using an Invitrogen Superscript III kit (#11752-050; Invitrogen) following the manufacturer's instructions to generate cDNA. cDNA was diluted to 15 ng/μl and primers to 2 μM stocks. Reactions were set up using 3.75 ng/μl DNA, 500 nM forward and reverse primers, and 2× DyNAmo Flash SYBR Green Master Mix (#F415; Thermo Fisher Scientific) in MicroAmp Optical 96-Well Reaction Plates (#N8010560; Applied Biosystems). RT-PCR was then performed on a QuantStudio 3 real-time PCR system using cycle: (1) 7 min at 95°C, (2) 15 s at 95°C, (3) 45 s at 60°C, and (4) 40 cycles of steps 2–3. RT-PCR was followed by melting curve analysis at 60–98°C for quality control. PCR products were tested on a gel for size during primer validation. RT-PCR results were analyzed with QuantStudio real-time PCR software.

Primer sequences are given in Table 3.

### Statistical analysis

All statistical analyses were performed with GraphPad Prism. Data were subjected to normality testing using the Shapiro–Wilk

Table 3. **Primer sequences**

| Target | Forward primer | Reverse primer | Reference |
|---|---|---|---|
| YWHAZ | 5′-AGGCATGTCTGTGTCCTAATG-3′ | 5′-CCTCCATAACCCATTCTTCT-3′ | Designed with IDT |
| BGN | 5′-GGGTCTCCAGCACCTCTACGC-3′ | 5′-TGAACACTCCCTTGGGCACCT-3′ | Fang et al. (2019) |
| FAM20C | 5′-GGCGGCCGTGGACTCCTATC-3′ | 5′-TTGACCATCCTGCCGGCCAC-3′ | Designed with Primer3 |
| PLOD1 | 5′-TGGAGGCTTCATCGGTTATG-3′ | 5′-GATGTAGTTGCCCAGGTAGTT-3′ | Designed with Primer3 |
| THSP1 | 5′-TTGGAACCACACCAGAAGAC-3′ | 5′-GACACTCAGTGCAGCTATCAA-3′ | Designed with Primer3 |
| ARHGDIB | 5′-ACCAATCACCATGGACCTTAC-3′ | 5′-AGGTGGTCTTGCTTGTCATC-3′ | Designed with Primer3 |
| SPARC | 5′-AATTCGGTCAGCTCAGAGTC-3′ | 5′-TGAGAAGGTGTGCAGCAATG-3′ | Designed with ExonSurfer |
| DCN | 5′-GCACTTTGTCCAGACCCAAATC-3′ | 5′-TGCTTGCACAAGTTTCCTGG-3′ | Designed with ExonSurfer |

test. Data found to be normally distributed were analyzed by t test or one-way ANOVA and the mean and standard deviation shown. Non-normally distributed data were analyzed with a Mann–Whiney test, or to compare multiple conditions, a Kruskal–Wallis test with Dunn's multiple comparisons were used, and the median and interquartile range were presented. Statistical information is provided in each figure legend.

### Online supplemental material

Fig. S1 details the genetic mutations in the GMAP210 and Golgin-160 RPE1 cell lines generated for this study and validation of KO. Fig. S2 provides additional data with respect to the ECM proteomics, namely, principal component analysis and collagen peptide analysis, as well as evidence that collagen processing is normal in KO cells. Fig. S3 contains immunofluorescence images showing that the early secretory pathway is unperturbed in KO cells. Fig. S4 provides principal component analysis and heat map depiction of secretome results. Fig. S5 shows steady-state BGN-SBP-mSc localization and line scans of RUSH movies showing cargo transit through the Golgi. Videos 1, 2, and 3 show 3D tomograms of Golgi membranes in WT and KO cells. Videos 4, 5, and 6 show live-imaging BGN transport through the Golgi in a RUSH assay in WT and KO cells. Videos 7, 8, and 9 show live imaging of SPARC transport through the Golgi in a RUSH assay in WT and KO cells.

### Data availability

The mass spectrometry proteomics data have been deposited to the ProteomeXchange Consortium (Deutsch et al., 2023) via the PRIDE partner repository (Vizcaíno et al., 2016) with the data set identifier PXD061302.

## Acknowledgments

We would like to thank Andrew Herman and Poppy Miller of the UoB FACS Facility with their help sorting CRISPR-transfected

cells, the UoB Wolfson Facility, especially Judith Mantell and Dominic Alibhai, for training advice and facilities, David Knight and Stacey Harwood at the University of Manchester BioMS Facility for running the matrisome samples, Prof. Oliver Jenson and Dr. Christopher Revell for conceptual discussions, Dr. Anne George and Dr. Harry Young for running pilot experiments, and Prof Stuart Haslam for advice.

This work was funded by the UK Research and Innovation–Biotechnology and Biological Sciences Research Council BB/T001984/1. Live imaging on the SpinSR system was supported by BBSRC Alert 19 equipment grant (BB/T017597/1). Open Access funding provided by University of Bristol.

Author contributions: G. Thompson: formal analysis, investigation, methodology, validation, visualization, and writing—review and editing. A. Hoyle: formal analysis, investigation, methodology, visualization, and writing—original draft. P.A. Lewis: formal analysis. M.E. Prada-Sanchez: investigation. J. Swift: funding acquisition, project administration, resources, supervision, and writing—review and editing. K. Heesom: investigation. M. Lowe: conceptualization, funding acquisition, supervision, and writing—review and editing. D. Stephens: conceptualization, funding acquisition, project administration, and supervision. N.L. Stevenson: conceptualization, data curation, formal analysis, funding acquisition, investigation, methodology, project administration, resources, supervision, validation, visualization, and writing—original draft, review, and editing.

Disclosures: The authors declare no competing interests exist.

Submitted: 22 November 2024

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

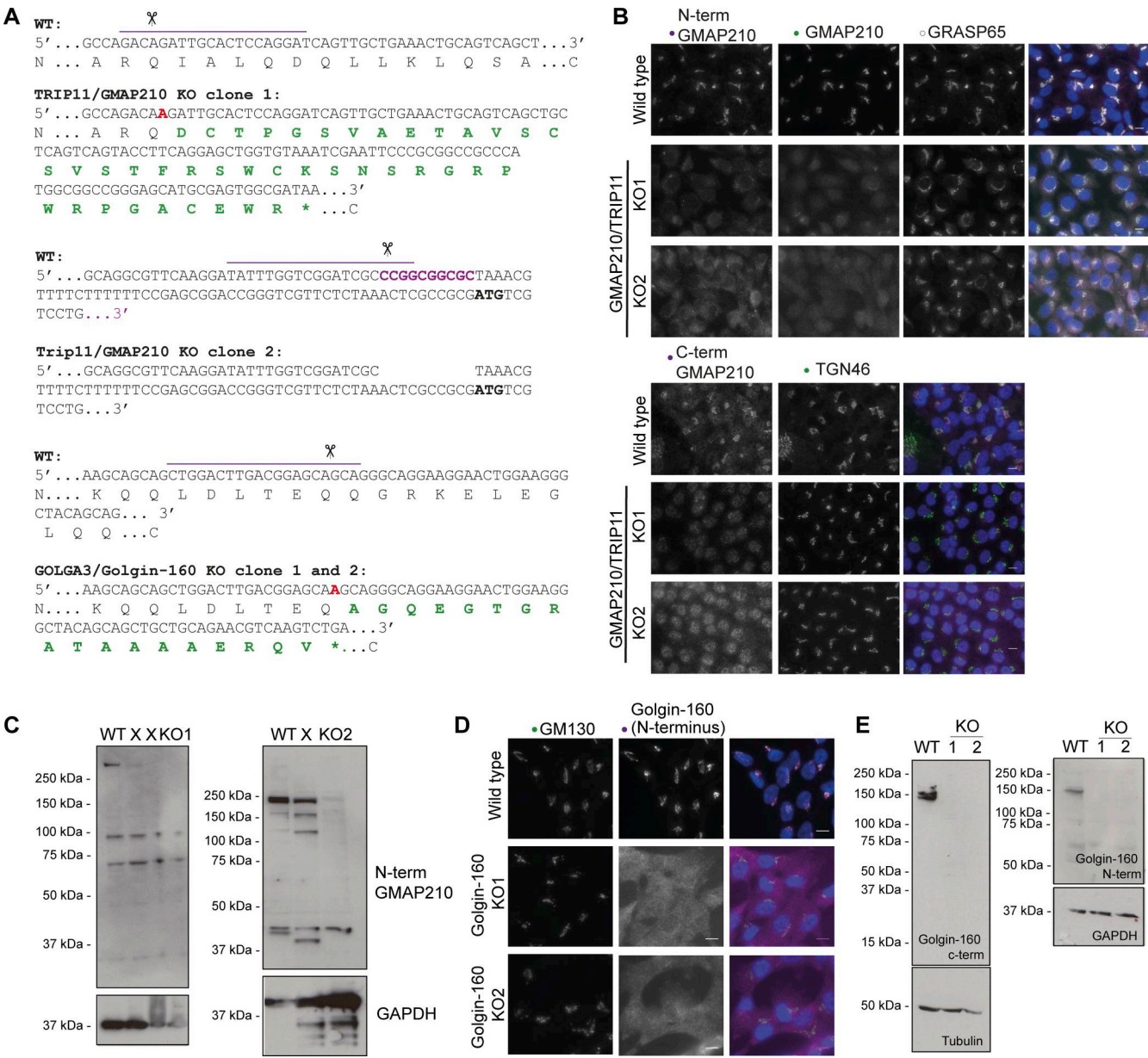

Figure S1. **Generation of CRISPR/Cas9 mutant lines. (A)** CRISPR design and resulting mutations in chosen KO clones. Top lines show gene sequence in WT and mutant (KO) clones with encoded amino acid sequence underneath. Purple lines indicate gRNA target sequence, scissors point to predicted Cas9 cut site, red letters in KO sequences indicate mutagenic base pair insertions, and purple letters in WT sequence indicate base pairs deleted in the mutant lines. Green amino acids are mutagenic changes arising after frameshift in KO lines, and * denotes a premature stop codon. **(B)** Maximum projection widefield images of WT and GMAP210 KO lines immunolabeled as indicated. GMAP210 antibodies were raised against amino acids 14–148 (N-term GMAP210), 159–365 (GMAP210), and 1760–1855 (C-term GMAP210). **(C)** Western blots of WT and GMAP210 KO cells probed with an antibody targeting amino acids 14–148 or GMAP210 (central well is an unsuccessful clone—X). **(D)** Maximum projection widefield images of WT and Golgin-160 KO clones labeled with GM130 (green, *cis*-Golgi) and Golgin-160 (magenta). **(D)** Western blot analysis of WT and Golgin-160 KO cell lysates probed with Golgin-160 antibodies and tubulin and GAPDH as loading controls. **(D and E)** Golgin-160 N-terminal and C-terminal antibodies raised against amino acids 1–350 and 1436–1498, respectively. Source data are available for this figure: SourceData FS1.

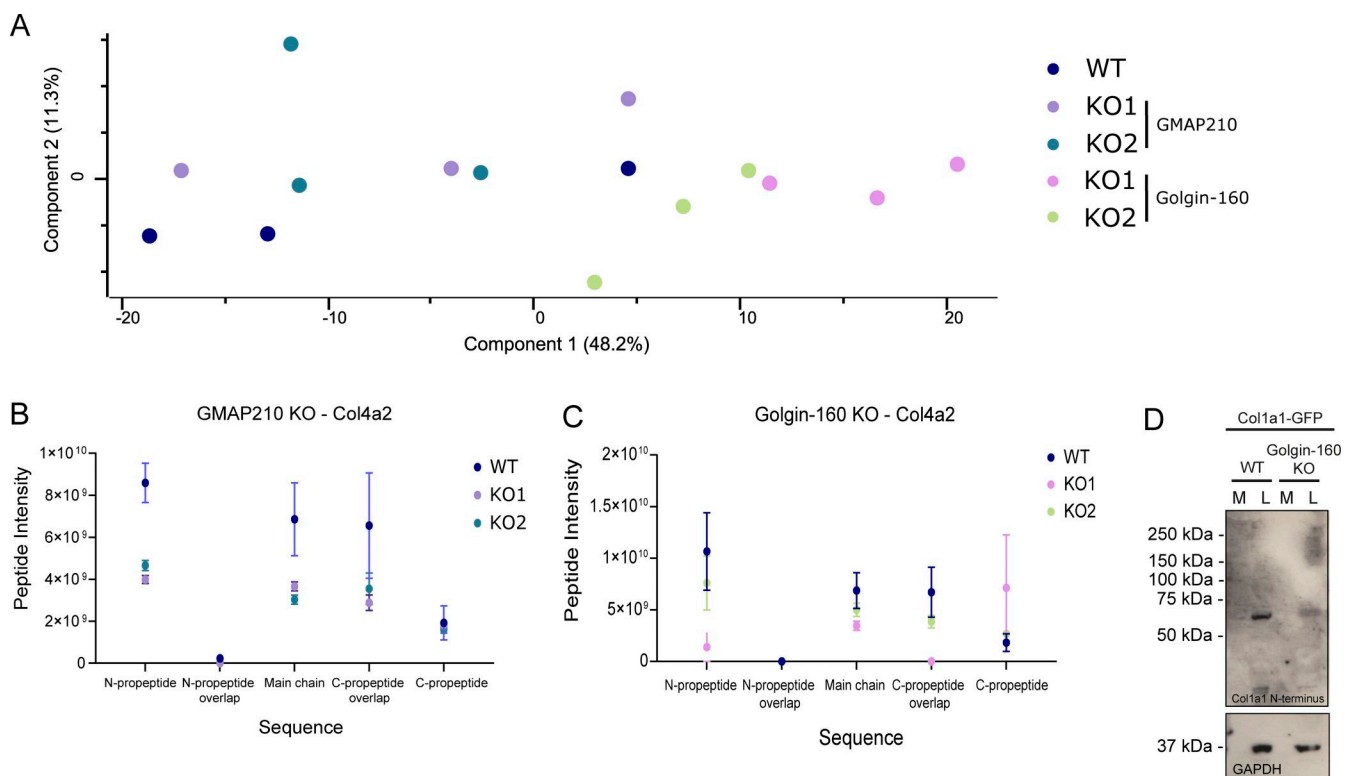

Figure S2. **Matrisome analysis in mutant cells. (A)** Principal component analysis of the mass spectrometry experiment shown in Fig. 3. **(B and C)** The Col4a2 sequence was split into its structural features, and abundance of relevant peptides was averaged within each feature and plotted for each mutant. Total Col4a2 abundance was normalized across conditions to investigate peptide-level variation. *N* = 3. **(D)** Western blots of medium (M) and lysate (L) fractions from WT and Golgin-160 KO cell cultures stably expressing pro-SBP-GFP-COL1A1. Blots probed with the LF39 antibody targeting the N-terminal propeptide domain of procollagen type I and GAPDH as a housekeeping protein. Source data are available for this figure: SourceData FS2.

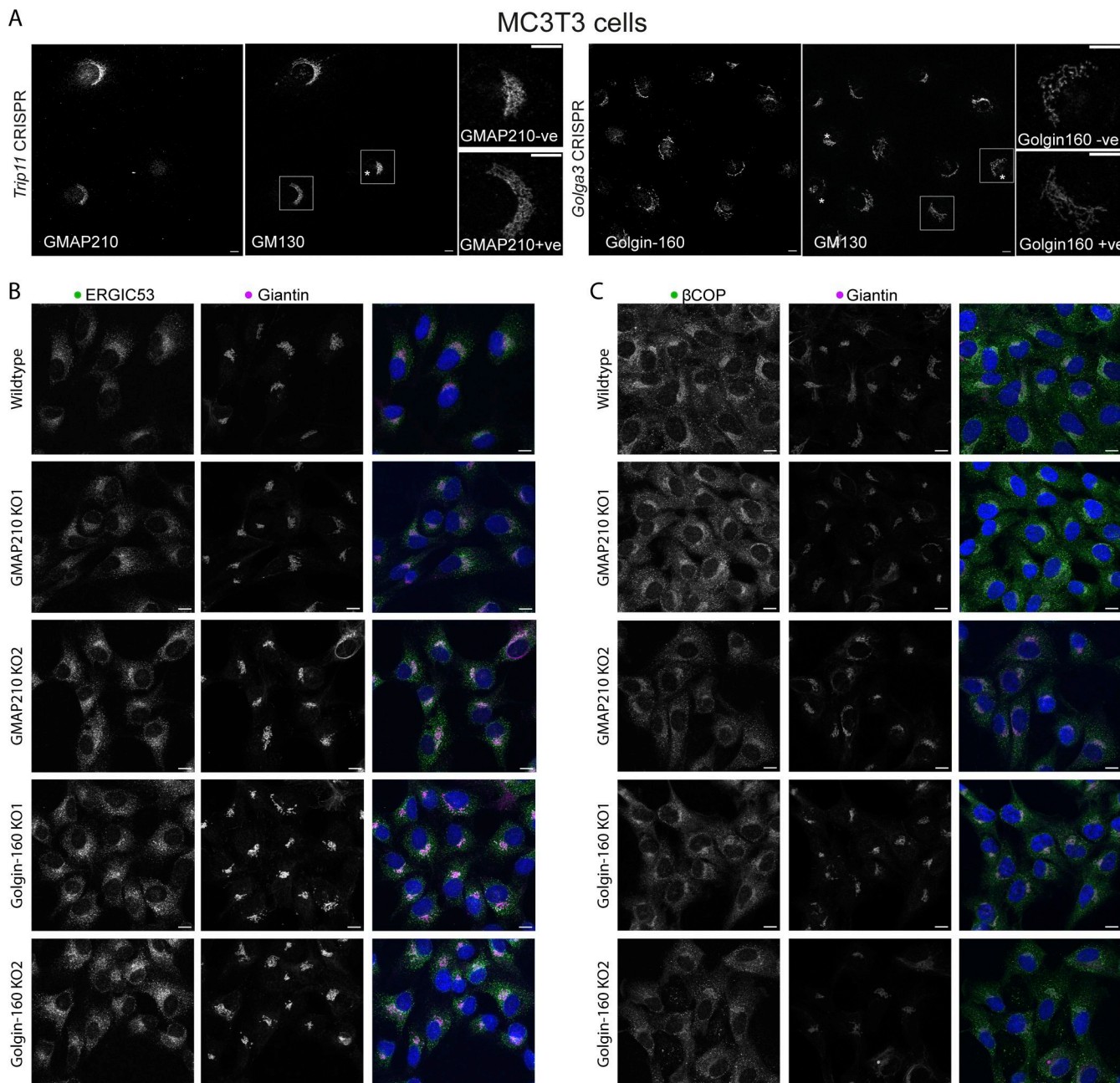

Figure S3. **Early secretory pathway organization in golgin mutants. (A)** Widefield images of MC3T3 cells transfected with Cas9 and gRNA targeting either *Trip11* or *Golga3* genes and stained for GMAP210 (Trip11) or Golgin-160 (Golga3) to identify KO cells (as indicated by an asterisk) and GM130. Inserts show GM130 label in a WT and a KO cell from a mixed population. Scale bar, 10 µM. **(B and C)** Confocal maximum projection images of WT and golgin KO cell lines stained for *cis/medial*-Golgi membrane (giantin, magenta) markers and either ERGIC (ERGIC53, green) (A) or COP1 (βCOP, green) structures (B). Scale bars, 10 µM.

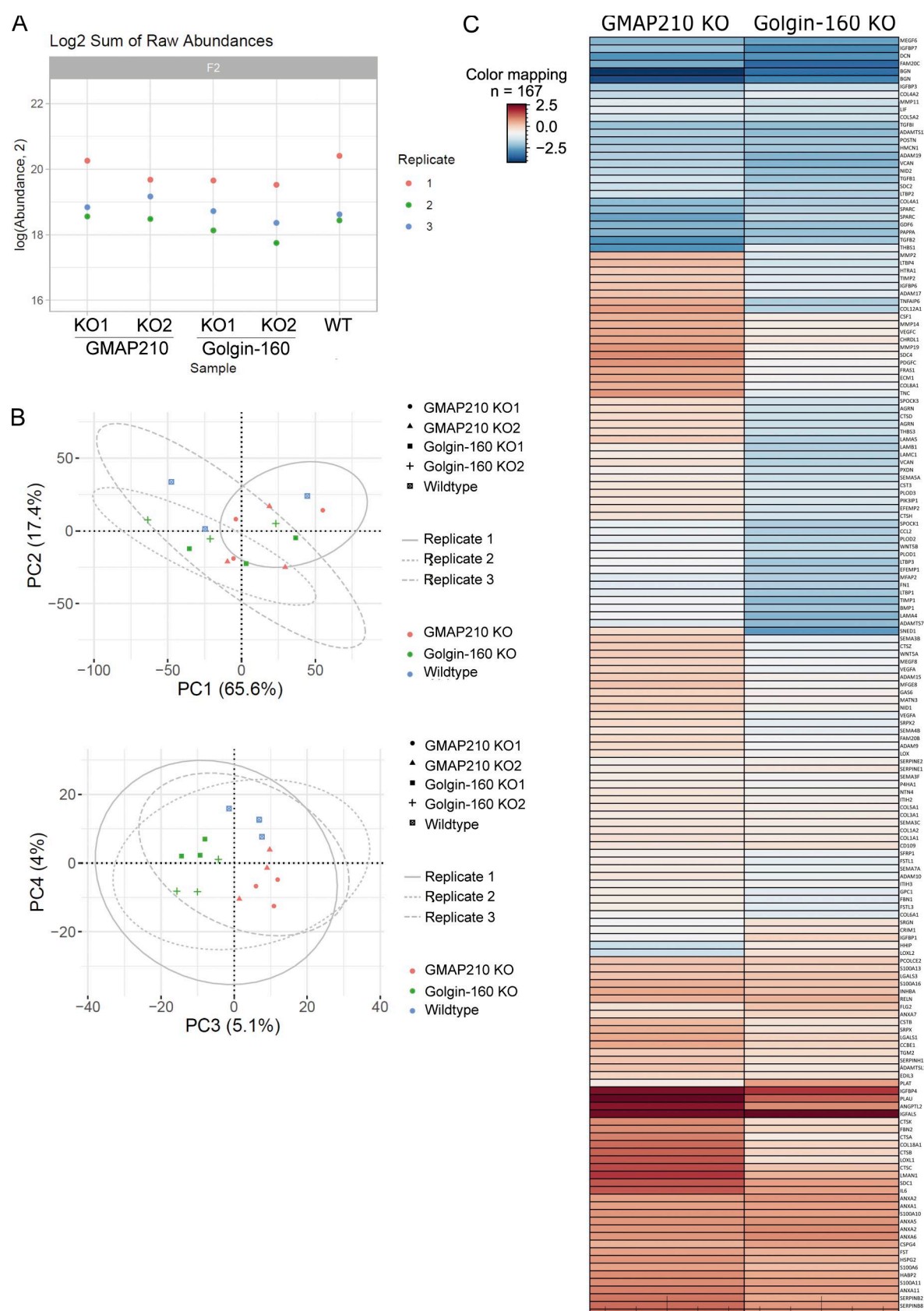

Figure S4. **Secretome analysis in mutant cells. (A)** Log2 sum of raw peptide abundance in each proteomics replicate from the secretome analysis shown in Fig. 5. **(B)** Principal component analysis of the secretome data sets represented in Fig. 5. **(C)** Heat map comparing protein abundance changes between the two different golgin KOs in the secretome experiment. Red and blue indicate increased and decreased abundance, respectively, and intensity of color is determined by the average log fold change across mutants compared with WT.

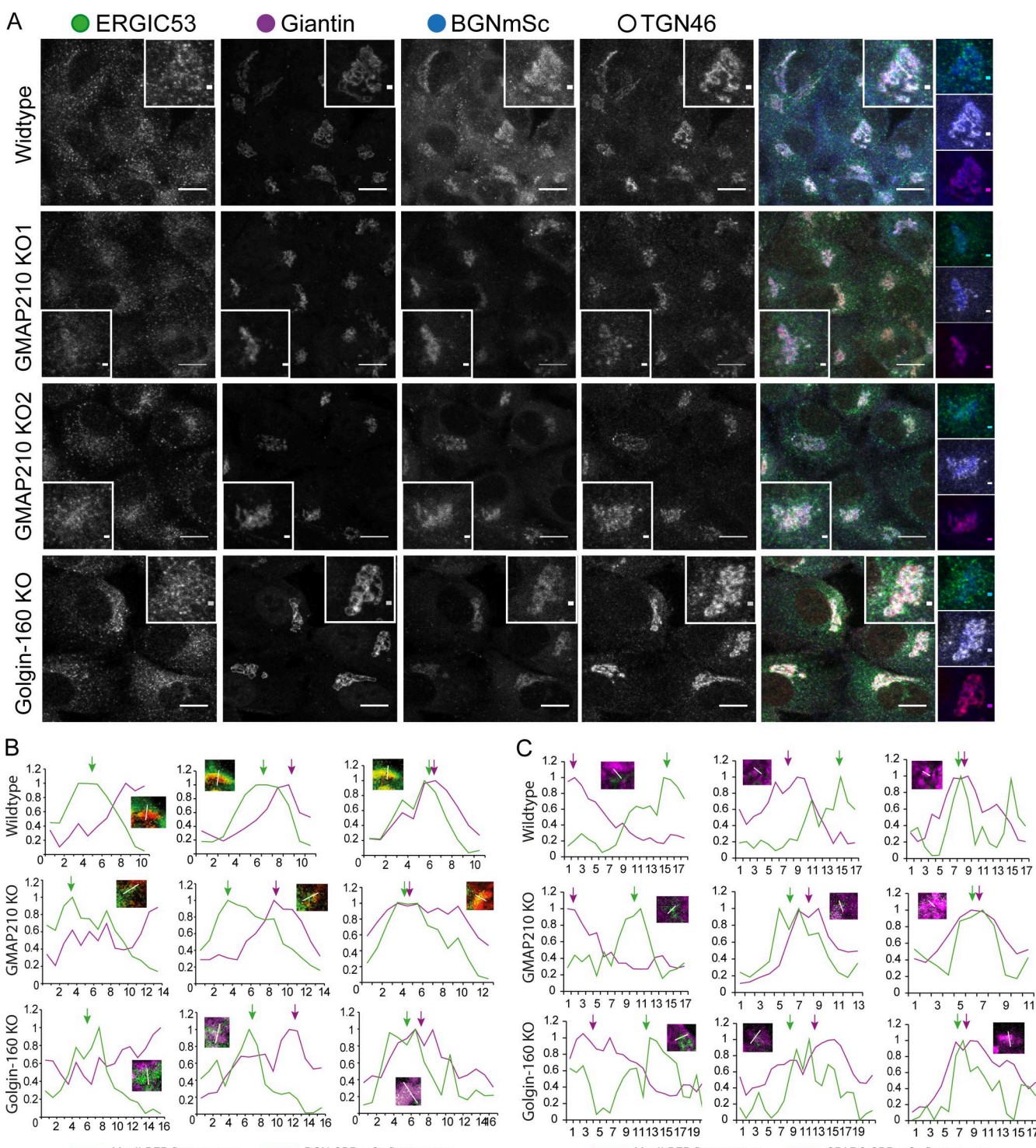

Figure S5. **BGN-SBP-mSc localization in mutant cells. (A)** Widefield maximum projections of cells stably expressing BGN-SBP-mSc (blue) and immunolabeled for the ERGIC (ERGIC53, green), *cis/medial*-Golgi (giantin, magenta), and TGN (TGN46, grayscale). Scale bar in main image, 10 µm; and insert, 1 µm. **(B and C)** Line-scan fluorescence intensity readings across Golgi elements measured at key time points in the RUSH assays shown in Figs. 6 and 7; and Videos 4, 5, 6, 7, 8, and 9. The region measured is shown in inset. Line traces show (B) BGN-SBP-mSc (magenta trace) or (C) SPARC-SBP-mSc accumulates adjacent to ManII-positive Golgi elements (green trace, middle column) and then colocalizes with ManII-BFP (right column). Fluorescence measurements are normalized to the maximal point of each trace to account for differential expression and photobleaching rates of each fluorescent protein. Arrows highlight peak for each protein. ManII, mannosidase II.

Video 1. **Golgi reconstruction and segmentation in WT cell.** Electron tomographic reconstruction of a 250-nm section through a WT cell showing an organized interconnected Golgi ribbon. Segmentation of a section of this ribbon shows cisternal membranes (labeled in blue/purple), dilated structures at the TGN (red), tubulovesicular structures (yellow), and vesicles (green). The movie shows slices running top–bottom–top through the section, then top to bottom adding the segmentation, followed by a zoom and rotational views of the 3D segmented structures. The movie runs at 40 frames per second and is also represented in Fig. 4. Scale bar, 500 nm.

Video 2. **Golgi reconstruction and segmentation in GMAP210 KO2 cell.** Electron tomographic reconstruction of a 250-nm section through a GMAP210 KO2 cell showing scattered and dilated Golgi structures. Segmentation of two closely apposed Golgi stacks shows cisternal membranes (labeled in blue/purple), dilated structures (red), tubulovesicular structures (yellow), and vesicles (green). The movie shows slices running top–bottom–top through the section, then top to bottom adding the segmentation, followed by a zoom and rotational views of the 3D segmented structures. The movie runs at 40 frames per second and is also represented in Fig. 4 F. Scale bar, 500 nm.

Video 3. **Golgi reconstruction and segmentation in the Golgin-160 KO2 cell.** Electron tomographic reconstruction of a 250-nm section through a Golgin-160 KO2 cell showing a fragmented Golgi, with stacks surrounded by vesicles. Segmentation of one stack shows cisternal membranes (labeled in blue/purple), dilated structures at the TGN (red), tubulovesicular structures (yellow), and vesicles (green). The movie shows slices running top–bottom–top through the section, then top to bottom adding the segmentation, then a zoom and rotational views of the 3D segmented structures. The movie runs at 40 frames per second and is also represented in Fig. 4 G. Scale bar, 500 nm.

Video 4. **BGN RUSH in a WT cell.** Spinning disk confocal time-lapse movie of a BGN RUSH assay in a WT cell. Cells are stably expressing BGN-SBP-mSc (magenta) and transiently expressing KDEL-streptavidin (unlabeled) and mannosidase II-BFP (green). The movie begins 15 min after 400 μM biotin addition (time after biotin addition is indicated in top left as mm:ss). One frame is taken every 500 ms. Images are single plane. The movie frame rate is 120 frames/s. Scale bar, 10 μm. The movie is also represented in Fig. 6 A.

Video 5. **BGN RUSH in a GMAP210 KO2 cell.** Spinning disk confocal time-lapse movie of a BGN RUSH assay in a GMAP210 KO2 cell. Cells are stably expressing BGN-SBP-mSc (magenta) and transiently expressing KDEL-streptavidin (unlabeled) and mannosidase II-BFP (green). The movie begins 15 min after 400 μM biotin addition (time after biotin addition is indicated in top left as mm:ss). One frame is taken every 500 ms. Images are single plane. The movie frame rate is 120 frames/s. Scale bar, 10 μm. The movie is also represented in Fig. 6 B.

Video 6. **BGN RUSH in a Golgin-160 KO2 cell.** Spinning disk confocal time-lapse movie of a BGN RUSH assay in a Golgin-160 KO2 cell. Cells are stably expressing BGN-SBP-mSc (magenta) and transiently expressing KDEL-streptavidin (unlabeled) and mannosidase II-BFP (green). The movie begins 10 min after 400 μM biotin addition (time after biotin addition is indicated in top left as mm:ss). One frame is taken every 500 ms. Images are single plane. The movie frame rate is 120 frames/s. Scale bar, 10 μm. The movie is also represented in Fig. 6 C.

Video 7. **SPARC RUSH in a WT cell.** Spinning disk confocal time-lapse movie of a SPARC RUSH assay in a WT cell. Cells are stably expressing SPARC-SBP-mSc (magenta) and transiently expressing KDEL-streptavidin (unlabeled) and mannosidase II-BFP (green). The movie begins 5 min after 400 μM biotin addition (time after biotin addition is indicated in top left as mm:ss). One frame is taken every 500 ms. Images are single plane. The movie frame rate is 120 frames/s. Scale bar, 10 μm. The movie is also represented in Fig. 7 A.

Video 8. **SPARC RUSH in a GMAP210 KO2 cell.** Spinning disk confocal time-lapse movie of a SPARC RUSH assay in a GMAP210 KO2 cell. Cells are stably expressing SPARC-SBP-mSc (magenta) and transiently expressing KDEL-streptavidin (unlabeled) and mannosidase II-BFP (green). The movie begins 5 min after 400 μM biotin addition (time after biotin addition is indicated in top left as mm:ss). One frame is taken every 500 ms. Images are single plane. The movie frame rate is 120 frames/s. Scale bar, 10 μm. The movie is also represented in Fig. 7 B.

Video 9. **SPARC RUSH in a Golgin-160 KO1 cell.** Spinning disk confocal time-lapse movie of a SPARC RUSH assay in a Golgin-160 KO1 cell. Cells are stably expressing SPARC-SBP-mSc (magenta) and transiently expressing KDEL-streptavidin (unlabeled) and mannosidase II-BFP (green). The movie begins 5 min after 400 μM biotin addition (time after biotin addition is indicated in top left as mm:ss). One frame is taken every 500 ms. Images are single plane. The movie frame rate is 120 frames/s. Scale bar, 10 μm. The movie is also represented in Fig. 7 C.

