## [Peer Review File · The Journal of Cell Biology]

Multiple golgins are required to support extracellular matrix secretion, modification and assembly

George Thompson, Anna Hoyle, Philip Lewis, M.Esther Prada-Sanchez, Joe Swift, Kate Heesom, Martin Lowe, David Stephens, and Nicola Stevenson

Corresponding Author(s): Nicola Stevenson, University of Bristol

Review Timeline:

Submission Date:	2024-11-22
Editorial Decision:	2025-01-21
Revision Received:	2025-06-09
Editorial Decision:	2025-06-30
Revision Received:	2025-07-04

Monitoring Editor: Sean Munro

Scientific Editor: Andrea Marat

Transaction Report:

DOI: <https://doi.org/10.1083/jcb.202411167>

January 21, 2025

Re: JCB manuscript #202411167

Nicola Stevenson
University of Bristol

Dear Dr. Stevenson,

Thank you for submitting your manuscript entitled "Golgi support extracellular matrix secretion by collectively maintaining the Golgi structure-function relationship". The manuscript was assessed by expert reviewers, whose comments are appended to this letter. We invite you to submit a revision if you can address the reviewers' key concerns, as outlined here.

You will see that the reviewers are overall positive about the quality and potential impact of your manuscript. Reviewer One has suggested some further experiments and modifications to clarify aspects of the work. Please address these where possible with existing or new data, or note if they are not technically achievable.

GENERAL GUIDELINES:

Text limits: Character count for an Article is < 40,000, not including spaces. Count includes title page, abstract, introduction, results, discussion, and acknowledgments. Count does not include materials and methods, figure legends, references, tables, or supplemental legends.

Figures: Articles may have up to 10 main text figures. Figures must be prepared according to the policies outlined in our Instructions to Authors, under Data Presentation, <https://jcb.rupress.org/site/misc/ifora.xhtml>. All figures in accepted manuscripts will be screened prior to publication.

*****IMPORTANT:** It is JCB policy that if requested, original data images must be made available. Failure to provide original images upon request will result in unavoidable delays in publication. Please ensure that you have access to all original microscopy and blot data images before submitting your revision. ***

Supplemental information: There are strict limits on the allowable amount of supplemental data. Articles may have up to 5 supplemental figures. Up to 10 supplemental videos or flash animations are allowed. A summary of all supplemental material should appear at the end of the Materials and methods section.

Please note that JCB now requires authors to submit Source Data used to generate figures containing gels and Western blots with all revised manuscripts. This Source Data consists of fully uncropped and unprocessed images for each gel/blot displayed in the main and supplemental figures. File names for Source Data figures should be alphanumeric without any spaces or special characters (i.e., SourceDataF#, where F# refers to the associated main figure number or SourceDataFS# for those associated with Supplementary figures). The lanes of the gels/blots should be labeled as they are in the associated figure, the place where cropping was applied should be marked (with a box), and molecular weight/size standards should be labeled wherever possible. Source Data files will be made available to reviewers during evaluation of revised manuscripts and, if your paper is eventually published in JCB, the files will be directly linked to specific figures in the published article.

The typical timeframe for revisions is three to four months. If you anticipate any difficulties in meeting this aforementioned revision time limit, please contact us and we can work with you to find an appropriate time frame for resubmission. Please note that papers are generally considered through only one revision cycle, so any revised manuscript will likely be either accepted or rejected.

Thank you for this interesting contribution to Journal of Cell Biology. You can contact us at the journal office with any questions at cellbio@rockefeller.edu.

Sincerely,

Sean Munro, PhD
Monitoring Editor

Andrea L. Marat, PhD
Deputy Editor

Journal of Cell Biology

Reviewer #1 (Comments to the Authors (Required)):

In this manuscript, the authors investigated the function of two specific golgins: GMAP-210 and Golgin-160, with a particular focus on their role in extracellular matrix (ECM) secretion. They report that inactivation of either protein leads to alterations in Golgi apparatus organization, albeit in slightly different manners: GMAP210 loss results in cisternal fragmentation and the accumulation of tubulovesicular structures, while Golgin-160 loss leads to fragmentation and vesicle accumulation. However, they report that both proteins affect ECM secretion and glycosaminoglycan synthesis, suggesting that these golgins, like Giantin, are essential for efficient ECM secretion, modification, and building a functional ECM. Such a function had not been reported before for Golgin-160.

Altogether it is an interesting study, reporting original data, in particular on the role of Golgin-160. There are still some points that may be answered before the study may be published.

Main points

1. When studying the "matrisome", the authors showed that many proteoglycans were affected, and their secretion was reduced. This may be due to impaired expression, altered transport secretion or degradation. Distinguishing whether the proteins were kept inside the cells, poorly expressed, or degraded would be important to understand the golgins' functions (this was partly studied for BGN).

2. Linked to this, my main point is that I did not understand why the authors chose to follow the impact of GMAP-210 and Golgin-160 KO on BGN transport and not on other proteoglycans. Indeed, while a reduction in BGN content in the matrix was observed, the authors also showed that BGN gene expression was reduced, which may explain the decline in its production. Unsurprisingly, no impact on trafficking was observed. Why not choose to study PLOD1 or THBS1, the secretion of which is reduced while no effect on expression was observed? This would have been more meaningful to test the role of these golgins on transport and release. I think at least one of these cargoes should be studied.

3. The effects observed on glycosylation are not very clear, as in many cases, it seems that the quantity of proteins rather than their status is affected. Particularly in Fig. 7A, the differences are not obvious. Loading less proteins and better adjusting the relative amount used in each condition might help (A loading control would be a nice addition actually). Also, differences in glycosylation are far clearer in conditions where the authors use NZ (Fig. 7F), even for control/untreated cells. If I understand correctly (it should be made clearer in the legend), one difference is that a 16-hour secretion (overnight) is shown in A, and a 5-hour secretion is analyzed in F. Perhaps the authors should reduce the time in A to get clearer results? Additionally, in the NZ experiment, it might have been better to pretreat the cells with NZ for at least an hour before starting the 5-hour secretion assay to ensure the Golgi was dispersed by NZ before starting the secretion assay. Also, in 7D, it is surprising that not all proteins get de-glycosylated upon PNGase treatment (including wild type). Overall, the effect of PNGase is not obvious.

4. The "compact" Golgi phenotype reported in Fig. 4A/B is not very clear (despite the quantification) and particularly when looking at Supplemental Fig. 1B or Supp 3. May be the authors should tune down a little bit their conclusions when commenting on the immunofluorescence data. The EM is far clearer. Also, why GM130 is not quantified there is unclear.

Minor points

5. When studying the transport of BGN, the authors report that the cargo initially accumulated adjacent to the ManII-GFP labeled structures. This seems particularly visible in GM210 KO cells, but it is far less visible under other conditions (including wild type). It would be interesting to quantify this for each cell lines.

6. The effect on ECM organization is clear from the images shown but it is unclear how the authors quantified it. Would be needed in the text or in the figure legend to mention the TWOMBLI plug-in and what is the principle behind analysis.

7. In Suppl Fig5, the Golgi in Golgin160 KO cells seem far more disorganized than in other figures, and the staining of BGN-mSc

is very weak as if these cells were not in good shape. May need to redo this experiment.

8. When commenting on the secreted proteins affected by the loss of GMAP210 and Golgin160, the authors report that a protein involved in osteoblast and chondrocyte differentiation was impacted. Was it expected that RPE-1 cells secrete these factors?

9. Quite a few defects seem linked to endo/lysosome proteins and even cytoplasmic protein release. Did the authors monitor exosomes or ECV release?

10. The authors should clearly indicate which cell line they are using in the text.

Reviewer #2 (Comments to the Authors (Required)):

The manuscript addresses the non-redundant roles of the two golgins GMAP210 and Golgins-160 using cell culture models. The authors developed novel knockout models for the cis-Golgi residing proteins and ask questions of their roles in Golgi structure and homeostasis. They have found that both are having impact on ECM organization and composition, while the poorly understood Golgin-160 leads to accumulation of small vesicles and fragmentation of Golgi ribbon. The authors use advance methods including CRISPR gene editing, high-resolution fluorescence microscopy, HS-AFM, and single-tilt electron tomography to discern Golgi structure and ECM organization, and multi-omics approaches (transcriptomics and proteomics with mass-spec imaging). This is a very clearly laid out and presented manuscript in the context of existing literature. The data are collected and analyzed in highly rigorous manner and proper statistical tools are applied.

The study makes several important points:

- a. GMAP210 and Golgins-160 are needed for fibrillar collagen assembly (using AFM), and KOs lead to reduced abundance in lysates.
- b. Mass spec data revealed non-redundant nature of the golgins in secretion of distinct proteins and organization of the ECM, also being distinct in comparison to giantin.
- c. The 3D tomography reconstructions revealed distinct disorganization of Golgi in both KOs, highlighting non-redundant roles of the two proteins residing in the same compartment. The authors show that GMAP210 is functioning in trafficking of ECM cargos and contributes to the architecture of the cis Golgi.
- d. Multiomics approaches show that shared and unique set of proteins and pathways are affected by individual KOs, but these are not necessarily concordant with transcriptional changes, suggesting functional regulation of secretory process.
- e. Analyses of the model cargo, biglycan, revealed very interesting requirement of both golgins in glycanation of the cargo.

Overall the choice of methods and experimental design is appropriate for the posed questions, and the conclusions are supported by the conducted experiments.

I congratulate authors on the clearly written and laid out mechanistic arguments shedding light on Cis-Golgi function and architecture. Couple of minor comments, it would be helpful to mention early in the results section the type of cells used for the KOs and the rationale for that selection. This information eventually can be reconstructed from different sections of the manuscript, but it comes late. Also, the Abstract section of the manuscript does not reflect the depth of the discovery presented in the manuscript, and it would benefit from a rewrite, with the focus on the new findings.

Reviewer #1 (Comments to the Authors (Required)):

In this manuscript, the authors investigated the function of two specific golgins: GMAP-210 and Golgin-160, with a particular focus on their role in extracellular matrix (ECM) secretion. They report that inactivation of either protein leads to alterations in Golgi apparatus organization, albeit in slightly different manners: GMAP210 loss results in cisternal fragmentation and the accumulation of tubulovesicular structures, while Golgin-160 loss leads to fragmentation and vesicle accumulation. However, they report that both proteins affect ECM secretion and glycosaminoglycan synthesis, suggesting that these golgins, like Giantin, are essential for efficient ECM secretion, modification, and building a functional ECM. Such a function had not been reported before for Golgin-160.

Altogether it is an interesting study, reporting original data, in particular on the role of Golgin-160. There are still some points that may be answered before the study may be published.

We thank the reviewer for their positive comments.

Main points

1. When studying the "matrisome", the authors showed that many proteoglycans were affected, and their secretion was reduced. This may be due to impaired expression, altered transport secretion or degradation. Distinguishing whether the proteins were kept inside the cells, poorly expressed, or degraded would be important to understand the golgins' functions (this was partly studied for BGN).

We thank the reviewer for raising this point. Our data show that BGN trafficking is normal, however its expression is downregulated, and believe this accounts for the 'matrisome' results for this particular cargo. We have now included qPCR data for a second proteoglycan, DCN showing that its expression is also downregulated (Figure 5D, line 312), suggesting this may be a broad mechanism explaining the reduced secretion of proteoglycans. One caveat for our trafficking conclusions is that we use an overexpression system to study BGN trafficking. We have attempted to study the endogenous protein to see if that is more revealing of the protein's fate however this was unsuccessful due to the low expression levels in the mutant cells. Antibody labelling proved to be too insensitive for reliable results. We also HiBit tagged the endogenous gene to quantify secretion by luciferase luminescence, which is more sensitive, however again only the WT cells produced detectable levels of endogenous protein. We agree that a more comprehensive study of the full proteoglycan family beyond SLRPs like BGN and DCN would be an interesting follow on from this study. Please also note points below RE SPARC and Golgi retention.

2. Linked to this, my main point is that I did not understand why the authors chose to follow the impact of GMAP-210 and Golgin-160 KO on BGN transport and not on other proteoglycans. Indeed, while a reduction in BGN content in the matrix was observed, the authors also showed that BGN gene expression was reduced, which may explain the decline in its production. Unsurprisingly, no impact on trafficking was observed. Why not choose to study PLOD1 or THBS1, the secretion of which is reduced while no effect on expression was observed? This would have been more meaningful to test the role of these golgins on transport and release. I think at least one of these cargoes should be studied.

We followed biglycan as it was the most significantly impacted cargo, however we do appreciate the reviewer's point about expression levels. At the time, we thought that this downregulation may be in response to a problem with trafficking or processing that could feedback upon transcription in an adaptive manner. Indeed, we did discover a defect in glycosylation, however cannot link this to expression at this stage.

As the reviewer has suggested we have now also investigated a second cargo. We agree that PLOD1 and THBS1 are excellent candidates, particularly as they do something different in each mutant and may reveal much about GMAP210 vs Golgin-160 function. However, because of this we also felt they were worthy of more in depth investigation outside of the scope of this study. We therefore instead chose SPARC, whose secretion is down-regulated in both mutants without any changes in expression (Figure 5C – new qPCR data, line 316-318). Interestingly, we found that SPARC takes an alternative ER-Golgi transport route in golgin KO cells, apparently using vesicular transport rather than a 'short-loop' pathway, indicative of adaptation to golgin loss (Figure 7 and supplemental movies 7-9). More importantly, Golgi transit/exit also appears impaired, with cargo being exchanged back and forth between Golgi elements rather than leaving in post-Golgi carriers. Indeed, Golgi retention was sufficiently severe in Golgin-160 KO that SPARC was clearly enriched in the Golgi at steady state (Figure 7E). This is sufficient to explain the reduced secretion reported in the proteomics. SPARC data is describe in lines 347-368 and construct design at lines 840-845.

3. The effects observed on glycosylation are not very clear, as in many cases, it seems that the quantity of proteins rather than their status is affected. Particularly in Fig. 7A, the differences are not obvious. Loading less proteins and better adjusting the relative amount used in each condition might help (A loading control would be a nice addition actually). Also, differences in glycosylation are far clearer in conditions where the authors use NZ (Fig. 7F), even for control/untreated cells. If I understand correctly (it should be made clearer in the legend), one difference is that a 16-hour secretion (overnight) is shown in A, and a 5-hour secretion is analyzed in F. Perhaps the authors should reduce the time in A to get clearer results?

We have now replaced the RFP blot in Figure 7A with another that is less exposed, and included a loading control. The figure legend has also been amended to include collection times.

Additionally, in the NZ experiment, it might have been better to pretreat the cells with NZ for at least an hour before starting the 5-hour secretion assay to ensure the Golgi was dispersed by NZ before starting the secretion assay.

We agree with the reviewer that a pre-incubation with nocodazole would have improved the experimental design for this assay, however this has proved to be technically challenging. An initial time course found that 5 hours was the maximal time we could comfortably incubate these cells in nocodazole without impacting cell adhesion and viability. We also found that 5 hours collection was the minimal time required for reliable and consistent results by western blot due to the amount of BGN secreted. Thus, we had to go with a 5-hour collection to balance these two limitations. We have previously shown that in this cell line, 90 minutes of nocodazole treatment was more than sufficient to fully disperse the Golgi (Stevenson et al JCS 2017) and thus we anticipate that at least 70% of the biglycan collected over 5 hours was processed in a fragmented Golgi. Whilst a preincubation would have made the results 'cleaner' and perhaps more striking if it were possible to do, we don't think it would have

changed our conclusion that nocodazole treatment reduces glycosylation and hope that the reviewer agrees.

Also, in 7D, it is surprising that not all proteins get de-glycosylated upon PNGase treatment (including wild type). Overall, the effect of PNGase is not obvious.

PNGase cleaves N-linked glycans by targeting GlcNAc bonded to asparagine, but leaves GAGs, which initiate through the addition of xylose to serine, intact. Current protein annotations indicate there are just two potential N-linked glycan sites on the BGN core protein at Asp 270 and Asp 311, and we predict their molecular weight to be substantially smaller than the two GAG chains present. Their relative contribution to the total molecular weight of the protein is therefore expected to be small and so we only see a small shift in MW upon their removal. As the samples were run on a gradient gel, these small changes in MW are better resolved for the smaller species upon PNGase treatment than for the higher molecular weight forms. We have added more arrows to help show the differences in glycosylated and deglycosylated forms that we hope help with interpretation of the blots.

4. The "compact" Golgi phenotype reported in Fig. 4A/B is not very clear (despite the quantification) and particularly when looking at Supplemental Fig. 1B or Supp 3. Maybe the authors should tune down a little bit their conclusions when commenting on the immunofluorescence data. The EM is far clearer. Also, why GM130 is not quantified there is unclear.

We acknowledge the reviewer's concerns here, however we do stand by our conclusions. Golgi structure between cells is always very heterogeneous and we have tried not to cherry pick extreme examples with our images, instead presenting the unbiased quantification that does show compaction. To our eye compacted Golgi can be seen in all figures, if not all cells, at the given resolution. To be sure, we have now performed a blinded qualitative analysis of images from WT and GMAP210 KO cells, with two team members sorting anonymised images into compacted and uncompact categories. Researchers successfully identified KO cells 70% of the time. During various experiments for another study we have also quantified a variety of Golgi markers of the cis- medial- and trans- Golgi and always see this trend (see below). We have also transfected a second cell line, MC3T3-E1, with Cas9 and gRNAs targeting GMAP210 and Golgin-160 and see the same phenotypes. This data has now been added to supplemental Figure 3 and is noted in line 203-204.

Minor points

5. When studying the transport of BGN, the authors report that the cargo initially accumulated adjacent to the ManII-GFP labeled structures. This seems particularly visible in GM210 KO cells, but it is far less visible under other conditions (including wild type). It would be interesting to quantify this for each cell lines.

We have found automated quantification of these movies challenging given the highly dynamic nature of the tagged proteins in a small area, alongside gradual photobleaching affecting signal-noise ratio over time. However, to better illustrate the particular point raised here we have been able to use line scans to look at spatial relations between the ManII and BGN or SPARC. These more clearly show cargo progression into the Golgi and have been added to Supplemental Figure 5.

6. The effect on ECM organization is clear from the images shown but it is unclear how the authors quantified it. Would be needed in the text or in the figure legend to mention the TWOMBLI plug-in and what is the principle behind analysis.

Use of the TWOMBLI plug-in is now stated in the figure legend and the methods section has been expanded to include the following explanation of the TWOMBLI plug in at lines 919-924: 'TWOMBLI combines the ridge detection and Anamorf plug ins to identify filamentous structures in images of extracellular matrix (Wershof et al., 2021). Masks are generated from identified filaments and various parameters are measured from these. Curvature is measured as the mean change in angle moving along a fibre, within a defined window. Alignment is computed by the plug in OrientationJ, which calculates the proportion of fibres within an image orientated in a similar direction'.

7. In Suppl Fig5, the Golgi in Golgin160 KO cells seem far more disorganized than in other figures, and the staining of BGN-mSc is very weak as if these cells were not in good shape. May need to redo this experiment.

The reviewer's observations are correct that this cell line does not tolerate the over-expressed cargo as well as controls. In the RUSH experiments, cells with lower expression levels were selected to ensure data were collected from healthy cells and some natural selection for lower expression also went on over time in the cultures, despite using similar gating parameters for FACS during the initial generation of all lines. The noted disorganisation of the Golgi is not dissimilar to that seen in a proportion of cells in the parental line (see comments above about Golgi heterogeneity) and is consistent with our

conclusions that loss of Golgin-160 causes Golgi fragmentation, however we concur the cell depicted displays a more extreme phenotype than the average cell. We have therefore now replaced this image with a healthier cell from the same experiment.

8. When commenting on the secreted proteins affected by the loss of GMAP210 and Golgin160, the authors report that a protein involved in osteoblast and chondrocyte differentiation was impacted. Was it expected that RPE-1 cells secrete these factors?

We think the reviewer is alluding to our GO term enrichment analysis which identified 7-10 proteins (see table 1) classified under the GO term osteoblast or chondrocyte differentiation. We agree that on the face of it this seems unusual however the proteins identified in this list tended to be ECM proteins rather than differentiation factors per se. These are important for osteoblasts and chondrocytes but not specific to them. The classification fell as follows:

Osteoblast differentiation	
ID	Gene Name
9188	DEAD-box helicase 21(DDX21)
1660	DEAD-box helicase 9(DHX9)
3020	H3.3 histone A(H3-3A)
3021	H3.3 histone B(H3-3B)
3491	cellular communication network factor 1(CCN1)
1277	collagen type I alpha 1 chain(COL1A1)
3192	heterogeneous nuclear ribonucleoprotein U(HNRNPU)
3486	insulin like growth factor binding protein 3(IGFBP3)
8482	semaphorin 7A (John Milton Hagen blood group)(SEMA7A)
1462	versican(VCAN)
Chondrocyte differentiation	
ID	Gene Name
81792	ADAM metallopeptidase with thrombospondin type 1 motif 12(ADAMTS1)
81029	Wnt family member 5B(WNT5B)
1490	cellular communication network factor 2(CCN2)
8091	high mobility group AT-hook 2(HMG2)
4054	latent transforming growth factor beta binding protein 3(LTBP3)
7040	transforming growth factor beta 1(TGFB1)
7045	transforming growth factor beta induced(TGFB1)
skeletal development	
ID	Gene Name
1277	collagen type I alpha 1 chain(COL1A1)
1290	collagen type V alpha 2 chain(COL5A2)
80781	collagen type XVIII alpha 1 chain(COL18A1)
10468	folliculin(FST)
7042	transforming growth factor beta 2(TGFB2)
1462	versican(VCAN)

9. Quite a few defects seem linked to endo/lysosome proteins and even cytoplasmic protein release. Did the authors monitor exosomes or ECV release?

We thank the reviewer for this interesting point, as it was not something we had considered. Our initial GO term analyses did not detect any enrichment for ECVs, however comparison our proteomics data with the exocarta database identified 31 of the top 100 exosomal proteins had a greater than 2-fold change in secretion in GMAP210 KO cultures when considering raw abundance ratios. Very few of these came out as statistically significant though after the two clones were pooled, which is likely why this was not picked out in the analysis, and why we have chosen not to add this in to the manuscript. Interestingly though, with the exception of the ECM protein FN1, none of these were present in the Golgin-160 dataset.

10. The authors should clearly indicate which cell line they are using in the text.

We thank the reviewer for highlighting this so we can improve the clarity of the manuscript. This has now been added to line 112.

Reviewer #2 (Comments to the Authors (Required)):

The manuscript addresses the non-redundant roles of the two golgins GMAP210 and Golgins-160 using cell culture models. The authors developed novel knockout models for the cis-Golgi residing proteins and ask questions of their roles in Golgi structure and homeostasis. They have found that both are having impact on ECM organization and composition, while the poorly understood Golgin-160 leads to accumulation of small vesicles and fragmentation of Golgi ribbon. The authors use advance methods including CRISPR gene editing, high-resolution fluorescence microscopy, HS-AFM, and single-tilt electron tomography to discern Golgi structure and ECM organization, and multi-omics approaches (transcriptomics and proteomics with mass-spec imaging). This is a very clearly laid out and presented manuscript in the context of existing literature. The data are collected and analyzed in highly rigorous manner and proper statistical tools are applied.

The study makes several important points:

- a. GMAP210 and Golgins-160 are needed for fibrillar collagen assembly (using AFM), and KOs lead to reduced abundance in lysates.
- b. Mass spec data revealed non-redundant nature of the golgins in secretion of distinct proteins and organization of the ECM, also being distinct in comparison to gigantín.
- c. The 3D tomography reconstructions revealed distinct disorganization of Golgi in both KOs, highlighting non-redundant roles of the two proteins residing in the same compartment. The authors show that GMAP210 is functioning in trafficking of ECM cargos and contributes to the architecture of the cis Golgi.
- d. Multiomics approaches show that shared and unique set of proteins and pathways are affected by individual KOs, but these are not necessarily concordant with transcriptional changes, suggesting functional regulation of secretory process.
- e. Analyses of the model cargo, biglycan, revealed very interesting requirement of both golgins in glycanation of the cargo.

Overall the choice of methods and experimental design is appropriate for the posed questions, and the conclusions are supported by the conducted experiments.

I congratulate authors on the clearly written and laid out mechanistic arguments shedding light on Cis-Golgi function and architecture.

We thank the reviewer for these positive and generous comments.

Couple of minor comments, it would be helpful to mention early in the results section the type of cells used for the KOs and the rational for that selection. This information eventually can be reconstructed from different sections of the manuscript, but it comes late.

We thank the reviewer for highlighting this so we can improve the clarity of the manuscript. The use of RPE1 cells has now been highlighted at the start of the results in line 112. These cells have previously been shown to produce a robust ECM and are conducive to the generation of long-term CRISPR lines (Stevenson et al 2021).

Also, the Abstract section of the manuscript does not reflect the depth of the discovery presented in the manuscript, and it would benefit from a rewrite, with the focus on the new findings.

We thank the reviewer for this feedback and the chance to improve the manuscript. We have now amended the abstract to be more explicit with respect to our findings. Note we have also had to change the title to adhere to JCB formatting.

June 30, 2025

RE: JCB Manuscript #202411167R

Nicola Stevenson
University of Bristol

Dear Dr. Stevenson:

Thank you for submitting your revised manuscript entitled "Multiple golgins are required to support extracellular matrix secretion, modification and assembly". We appreciate your thoughtful responses to the reviewers and that you have added a substantial amount of new data. Overall, we agree that you have successfully addressed their comments and suggestions and would be happy to publish your paper in JCB pending final revisions necessary to meet our formatting guidelines (see details below).

A. MANUSCRIPT ORGANIZATION AND FORMATTING:

1) Text limits: Character count for Articles is < 40,000, not including spaces. Count includes abstract, introduction, results, discussion, and acknowledgments. Count does not include title page, figure legends, materials and methods, references, tables, or supplemental legends.

2) Figures limits: Articles may have up to 10 main text figures.

3) Figure formatting: Scale bars must be present on all microscopy images, including inset magnifications. Molecular weight or nucleic acid size markers must be included on all gel electrophoresis. Aspect ratios of images may not be altered.

* In addition:

a) In figure 7, although the main text and the figure legend refers to SPARC, the labelling on the figure says "Biglycan". Please correct.

b) The legend to Figure 7 refers to a panel F which is not on the figure.

c) As per JCB policy please add scale bars to the high quality EM shown in Supplemental Movies 1-3.

4) Statistical analysis: Error bars on graphic representations of numerical data must be clearly described in the figure legend. The number of independent data points (n) represented in a graph must be indicated in the legend. Statistical methods should be explained in full in the materials and methods. For figures presenting pooled data the statistical measure should be defined in the figure legends. Please also be sure to indicate the statistical tests used in each of your experiments (either in the figure legend itself or in a separate methods section) as well as the parameters of the test (for example, if you ran a t-test, please indicate if it was one- or two-sided, etc.). Also, if you used parametric tests, please indicate if the data distribution was tested for normality (and if so, how). If not, you must state something to the effect that "Data distribution was assumed to be normal but this was not formally tested."

5) Abstract and title: The abstract should be no longer than 160 words and should communicate the significance of the paper for a general audience. The title should be less than 100 characters including spaces. Make the title concise but accessible to a general readership.

6) Materials and methods: Should be comprehensive and not simply reference a previous publication for details on how an experiment was performed. Please provide full descriptions in the text for readers who may not have access to referenced manuscripts.

7) All antibodies, cell lines, animals, and tools used in the manuscript should be described in full, including accession numbers for materials available in a public repository such as the Resource Identification Portal. Please be sure to provide the sequences for all of your primers/oligos and RNAi constructs in the materials and methods. You must also indicate in the methods the source, species, and catalog numbers (where appropriate) for all of your antibodies. Please also indicate the acquisition and quantification methods for immunoblotting/western blots.

8) Microscope image acquisition: The following information must be provided about the acquisition and processing of images:

- a. Make and model of microscope
- b. Type, magnification, and numerical aperture of the objective lenses
- c. Temperature
- d. Imaging medium
- e. Fluorochromes
- f. Camera make and model
- g. Acquisition software
- h. Any software used for image processing subsequent to data acquisition. Please include details and types of operations involved (e.g., type of deconvolution, 3D reconstitutions, surface or volume rendering, gamma adjustments, etc.).

10) Supplemental materials: There are strict limits on the allowable amount of supplemental data. Articles may have up to 5 supplemental figures. Please also note that tables, like figures, should be provided as individual, editable files. A summary of all supplemental material should appear at the end of the Materials and methods section.

13) ORCID IDs: ORCID IDs are unique identifiers allowing researchers to create a record of their various scholarly contributions in a single place. Please note that ORCID IDs are now *required* for all authors. At resubmission of your final files, please be sure to provide your ORCID ID and those of all co-authors.

Please note that JCB now requires authors to submit Source Data used to generate figures containing gels and Western blots with all revised manuscripts. This Source Data consists of fully uncropped and unprocessed images for each gel/blot displayed in the main and supplemental figures. For assays performed using capillary electrophoresis and/or immunoassay-based detection, authors should instead provide the electropherogram graph(s) for each experiment, plotting fluorescence/chemiluminescence intensity vs. molecular weight/size. Please be sure to provide one Source Data file for each figure gels, blots, and/or capillary electrophoresis assays along with your revised manuscript files. File names for Source Data figures should be alphanumeric without any spaces or special characters (i.e., SourceDataF#, where F# refers to the associated main figure number or SourceDataFS# for those associated with Supplementary figures). For traditional gels and blots, the lanes of the gels/blots should be labeled as they are in the associated figure, the place where cropping was applied should be marked (with a box), and molecular weight/size standards should be labeled wherever possible. For capillary electrophoresis assays, each trace in the graph should be color-coded and labeled to indicate which protein, gene, or sample is being measured (please try to avoid red/green combinations to accommodate our color-blind readers).

Journal of Cell Biology now requires a data availability statement for all research article submissions. These statements will be published in the article directly above the Acknowledgments. The statement should address all data underlying the research presented in the manuscript. Please visit the JCB instructions for authors for guidelines and examples of statements at (<https://rupress.org/jcb/pages/editorial-policies#data-availability-statement>).

B. FINAL FILES:

****It is JCB policy that if requested, original data images must be made available to the editors. Failure to provide original images upon request will result in unavoidable delays in publication. Please ensure that you have access to all original data images prior to final submission.****

****The license to publish form must be signed before your manuscript can be sent to production. A link to the electronic license to publish form will be sent to the corresponding author only. Please take a moment to check your funder requirements before choosing the appropriate license.****

Thank you for your attention to these final processing requirements. Please revise and format the manuscript and upload materials within 7 days. If you need an extension for whatever reason, please let us know and we can work with you to determine a suitable revision period.

Thank you for this interesting contribution, we look forward to publishing your paper in Journal of Cell Biology.

Sincerely,

Sean Munro, PhD
Monitoring Editor

Andrea L. Marat, PhD
Deputy Editor

Journal of Cell Biology